# An Empirical Study of Pre-trained Model Selection for Out-of-Distribution Generalization and Calibration

**Hiroki Naganuma**[1,2]†**, Ryuichiro Hataya**[3]†**, Kotaro Yoshida**[4]**, Ioannis Mitliagkas**[1,2]
*naganuma.hiroki@mila.quebec, ryuichiro.hataya@riken.jp, yoshida.k.0253@m.isct.ac.jp, ioannis@mila.quebec,*
[1]*Mila - Quebec AI Institute,* [2]*Université de Montréal,* [3]*RIKEN AIP,* [4]*Institute of Science Tokyo,*

**Reviewed on OpenReview:** *https://openreview.net/forum?id=tYjoHjShxF*

## Abstract

In the field of computer vision, fine-tuning pre-trained models has become a prevalent strategy for out-of-distribution (OOD) generalization tasks. Different from most prior work that has focused on advancing learning algorithms, we systematically examined how pre-trained model size, pre-training dataset size, and training strategies impact generalization and confidence calibration on downstream tasks. We evaluated 100 models across diverse pre-trained model sizes, five pre-training datasets, and five data augmentations through extensive experiments on four distribution shift datasets totaling over 120,000 GPU hours. Our results demonstrate the significant impact of pre-trained model selection, with optimal choices substantially improving OOD accuracy over algorithm improvement alone. Additionally, we find that larger models and bigger pre-training datasets not only enhance OOD performance but also improve calibration, helping to mitigate overconfidence, contrary to some prior studies that found modern deep networks to calibrate worse than classical shallow models. Our work underscores the overlooked importance of pre-trained model selection for out-of-distribution generalization and calibration.

## 1 Introduction

Out-of-distribution (OOD) generalization for computer vision models has become critical challenges in recent years. This stems from their indispensable role in developing robust and reliable artificial intelligence across diverse tasks and situations (Nagarajan et al., 2021; Geirhos et al., 2020; Shen et al., 2021; Zhou et al., 2021). Our research holds particular relevance to practical contexts requiring OOD generalization in computer vision, such as domain generalization tasks (Arjovsky et al., 2019b; Ghifary et al., 2015; Li et al., 2017; Fang et al., 2013; Venkateswara et al., 2017; Peng et al., 2019).

Currently, the prevalent strategy for tackling this issue relies on fine-tuning—adjusting the parameters of pre-trained models to optimize performance on specified downstream tasks (Gulrajani & Lopez-Paz, 2021).

Table 1: Comparison of OOD test accuracy of ResNet-50 trained with ERM (Vapnik, 1991), ResNet-50 with the best algorithms (Best Alg.) reported in Gulrajani & Lopez-Paz (2021), and a larger model (ViT-base pre-trained on ImageNet-21k for VLCS and ViT-large pre-trained on ImageNet-21k for other tasks) trained with ERM. Δ indicates the improvement from results with the best algorithms to results with the best models. Baseline results are adopted from Gulrajani & Lopez-Paz (2021).

|  | ERM | Best Alg. | Best Model | Δ |
|---|---|---|---|---|
| PACS | 88.1 | 89.1 | 96.7 | +7.71 |
| OfficeHome | 62.7 | 64.7 | 87.4 | +22.7 |
| DomainNet | 58.4 | 59.5 | 77.8 | +18.3 |
| VLCS | 76.4 | 77.5 | 82.2 | +4.7 |

---

*These authors contributed equally to this work.

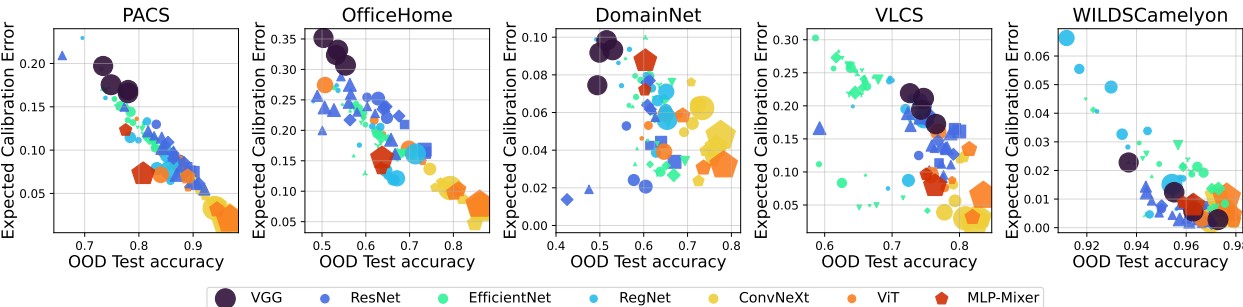

Figure 1: This paper investigates the out-of-distribution (OOD) generalization and confidence calibration of a total of 100 ImageNet pre-trained models on five datasets from DomainBed (Gulrajani & Lopez-Paz, 2021) and WILDS (Koh et al., 2021). These panels show the relationship between expected calibration error (ECE) rates (lower is better) and OOD test accuracy (higher is better). Marker sizes are proportional to the number of model parameters, and marker shapes correspond to pre-training configurations. We can observe a general trend that **larger models achieve the best of both worlds (bottom-right corner)**, except for VGGs and MLP-Mixers. *The full legend can be found in fig. D.6.*

A limitation of current approaches is that existing research has predominantly focused on advancing learning algorithms (Arjovsky et al., 2019a; Rame et al., 2022; Krueger et al., 2021; Sun & Saenko, 2016; Sagawa et al., 2019; Yan et al., 2020; Blanchard et al., 2021; Koyama & Yamaguchi, 2020; Ahuja et al., 2021), with limited investigation into how the choice of pre-trained model affects OOD generalization. More recent studies (Li et al., 2022; Cha et al., 2022; Arpit et al., 2022; Zhang et al., 2023) have evaluated their proposed methodology on more modern architectures, like vision transformers; however, they also used a limited number of pre-trained models solely to benchmark the effectiveness of their proposed OOD generalization algorithms. While this approach has yielded significant progress, it often overlooks the crucial impact of pre-trained model selection on OOD generalization.

Recently, some works (Goldblum et al., 2023; Vishniakov et al., 2023; Zhang et al., 2022) have studied the effects of pre-trained model selection on OOD generalization; however, these papers partially answer a subset of the questions that we pose, and leave other important questions unanswered. Specifically, Goldblum et al. (2023) evaluates several model architectures and model sizes as backbones in various downstream tasks but does not evaluate and discuss the scaling laws in model size and dataset size that we found. Zhang et al. (2022) focuses exclusively on Vision Trainformer (ViT) models and, Vishniakov et al. (2023) focuses on both ViT and ConvNext models. Our experiments are designed to answer the important questions left unanswered by these studies. What is the effect of dataset size, and the relevant scaling laws? Do the phenomena on which we focus persist when we study a wide class of architectures?

In section 3, we provide more details on the related work.

Motivated by these limitations, we conduct a comprehensive and systematic evaluation of pre-trained model selection for OOD generalization. We have found that pre-trained model selection is more powerful than initially thought, leading to substantial performance improvements over algorithmic enhancements (See table 1).

Furthermore, to gain a deeper understanding of the scaling laws observed in OOD generalization, we also evaluated the models from the perspective of confidence calibration (Naeini et al., 2015; Guo et al., 2017; Krishnan & Tickoo, 2020), which aims to ensure that the estimated class probabilities are consistent with actual class probabilities. Calibration has been shown to have a strong connection to OOD generalization (Wald et al., 2021; Ovadia et al., 2019; Immer et al., 2021; Naganuma & Kimura, 2023; Yoshida & Naganuma, 2024), and it allows for a more detailed analysis by focusing on the model's confidence, which cannot be captured by accuracy alone. Similar to OOD generalization, research on calibration in pre-trained models has been limited to specific settings, and a generalized understanding has yet to be established.

We evaluate pre-trained models along three dimensions: (1) pre-trained model parameter size and architecture, (2) pre-training dataset size, (3) model design and training strategy for pre-trained models. We assess

performance on downstream tasks using two key metrics: accuracy and calibration errors under distributional shift. For calibration, we employ Expected Calibration Error (ECE) (Naeini et al., 2015; Nixon et al., 2019). The few existing studies for this setting do not thoroughly explore all three axes above, an exploration that is important for achieving the best results from model selection and for investigating this factor.

Through extensive experiments on five distribution shift datasets, we provide new insights into pre-trained model selection for OOD generalization in computer vision. Specifically, we evaluated 100 pre-trained models, spanning five pre-training datasets (ImageNet-1k to 22k and JFT-300M (Sun et al., 2017)), five data augmentation strategies, and several pre-training algorithms (including DINO (Caron et al., 2021), MAE (He et al., 2022), AdvProp (Xie et al., 2020a), Noisy Student (Xie et al., 2020b)) and CLIP (Radford et al., 2021) on downstream tasks over five distribution shift datasets. Our extensive experiments, involving over 120,000 GPU hours, consistently show that larger models and more extensive datasets improve OOD generalization and calibration. As can be seen in table 1, compared to learning algorithm improvements (up to 2% on DomainBed (Gulrajani & Lopez-Paz, 2021)), the best pre-trained model selection boosts OOD accuracy dramatically (up to 22.7% in our experiments). These results underscore that pre-trained model selection plays a critical yet overlooked role in OOD generalization.

**Contributions**

This work highlights three key insights into model selection for OOD generalization:

- We perform a broad-scope study which establishes that larger model sizes and bigger pre-training datasets improve OOD generalization (fig. 3). Our study is the first to evaluate the effect of dataset size, and model size on a wide set of architectures, and it lends solid experimental support to common intuition among researchers.

- A deeper analysis of OOD generalization through the lens of calibration reveals that scaling laws also hold for calibration. This suggests that the signs of scaling laws observed in OOD generalization can be explained, in part, from the perspective of calibration within the context of invariant feature learning (fig. 3).

- The generic model design and training strategy produces the most transferable pre-trained models, compared to techniques that overfit on ImageNet-1K (fig. 4 and fig. 5).

Our systematic findings advocate an increased focus on pre-trained model selection. Specifically, we highlight best practices like using the largest pre-trained models and datasets and avoiding techniques tailored narrowly for in-distribution performance. These insights could redefine approaches to OOD generalization.

## 2 Preliminary

Before delving into our experiments and findings, it is crucial to lay the groundwork by defining key concepts that underpin our research.

**OOD Generalization:** Out-of-distribution (OOD) generalization encompasses a range of concepts, including domain generalization, domain adaptation, robust generalization, and adversarial training. It aims to represent a model's capacity to accurately interpret and handle data that it has not encountered during training. This aspect is crucial for ensuring that models perform effectively in real-world scenarios where data can vary from the training set distribution.

**Confidence Calibration:** Quantifying the correctness of model's confidence is vital for understanding the reliability of the model prediction. This becomes particularly relevant in high-stakes domains like cancer detection, where a model's overconfidence can have serious consequences. Guo et al. (2017) showed that "modern" deep neural networks (in 2017), such as ResNet-110, tend to be overconfident in their predictions compared to classical neural networks such as LeNet-5, highlighting the need for reliable quantification of confidence calibration.

Expected Calibration Error (ECE) is a widely used metric for quantifying confidence calibration. It measures the discrepancy between a model's predicted confidence and its actual accuracy. The formula is given as:

$$\text{ECE} = \sum_{m=1}^{M} \frac{|B_m|}{n} \left| \text{acc}\left(B_m\right) - \text{conf}\left(B_m\right) \right|. \tag{1}$$

Accuracy $\text{acc}\left(B_m\right)$ and confidence $\text{conf}\left(B_m\right)$ are defined as follows

$$\text{acc}\left(B_m\right) = \frac{1}{|B_m|} \sum_{i \in B_m} \mathbf{1}\left(\hat{y}_i = y_i\right), \quad \text{conf}\left(B_m\right) = \frac{1}{|B_m|} \sum_{i \in B_m} \hat{p}_i, \tag{2}$$

where $B_m$ is the set of samples whose prediction scores fall into bin $m$, $\hat{p}_i$ is the confidence of sample $i$, $y_i$ is the true label, $\hat{y}_i$ is the predicted label, and $n$ is the total number of samples in all the bins.

While ECE provides a useful overall measure of calibration, it is important to interpret ECE carefully, especially when accuracy is low. Models that make random predictions or fail to learn can still achieve low ECE. Therefore, when comparing models, it is critical to consider ECE primarily in high-accuracy regions. Furthermore, ECE suffers from vulnerability to biases in model confidence due to its use of fixed confidence bins. Specifically, recent large-scale neural networks are typically highly confident, resulting in a substantial number of data points concentrated in high-confidence bins while low-confidence bins contain very few data points. This imbalance leads to variability in the accuracy of the estimated errors. To address this issue, Nixon et al. (2019) proposed ACE. ACE adaptively partitions the confidence scores into bins such that each bin contains an equal number of instances. It is defined by the following equation:

$$\text{ACE} = \frac{1}{KM} \sum_{k=1}^{K} \sum_{m=1}^{M} \left| \text{acc}\left(B_{m,k}\right) - \text{conf}\left(B_{m,k}\right) \right|, \tag{3}$$

where $K$ denotes the number of classification classes, and $B_{m,k}$ represents the $m$-th bin within the model's confidence for class $k$. For each class, the model confidence are sorted, and the binning is performed such that each bin contains $\lfloor N/M \rfloor$ data points evenly.

## 3 Related Works

**OOD Generalization Algorithms and Benchmarks:** In recent years, various algorithms have been proposed to improve OOD generalization, such as invariant risk minimization (Arjovsky et al., 2019a), group distributionally robust optimization (Sagawa et al., 2019), risk extrapolation (Krueger et al., 2021), correlation alignment (Sun & Saenko, 2016), gradient variances control (Rame et al., 2022). However, their performance depends heavily on the dataset characteristics, and no optimal algorithm has been identified. Furthermore, empirical risk minimization still achieves competitive performance on covariate shift (Ahuja et al., 2020). DomainBed (Gulrajani & Lopez-Paz, 2021) has become a standard benchmark consisting of image classification datasets with different domain shifts for evaluating OOD generalization. Other benchmarks have also been introduced, such as WILDS (Koh et al., 2021) for various data types and distributional shifts, WOODS (Gagnon-Audet et al., 2022) focusing on time series data, and TableShift (Gardner et al., 2023) for tabular data.

There are four differences between their studies and ours. First, our protocol targets pre-trained model selection for fine-tuning tasks, whereas their protocol targets scratch training. Second, we evaluate in the context of OOD generalization, and they evaluate on robustness benchmarks such as image corruption and perturbation (We also provide the results in fig. D.9, in addition to the OOD generalization benchmarks). Third, we evaluate more than 100 models, including a wide range of architectures, training strategies, model parameter size, and training data, as well as an exhaustive hyperparameter search, which has been done with a limited number of model families and training sets (28 at most) in the previous study. Finally, we conduct a systematic evaluation of the impact of the number of parameters in the pre-trained model and the size of the pre-training dataset on OOD generalization, with a deeper analysis also examining calibration metrics. This is an attempt to explore an aspect that has not been previously investigated.

**Pre-trained Model Selection for Downstream Tasks:** Fine-tuning pre-trained models has become the prevalent approach for OOD generalization tasks (Gulrajani & Lopez-Paz, 2021). However, most works have focused on advancing algorithms, with a limited investigation into pre-trained model selection. Some analyses (Goldblum et al., 2023; Zhang et al., 2022; Tada & Naganuma, 2023; Yamada & Otani, 2022; Andreassen et al., 2021) have compared model architectures on robustness benchmarks (Hendrycks et al., 2021a;b; Beyer et al., 2020; Recht et al., 2019; Hendrycks & Dietterich, 2019).

However, in studies that measured the effect of pre-trained model selection for OOD generalization, the number of pre-trained models for comparison is limited. For example, Carlucci et al. (2019) proposed pre-training CNNs to solve puzzles to learn features useful for domain generalization and evaluated them using ResNet18 and AlexNet as pre-trained models. Li et al. (2022) proposed a GMoE model that replaces the FFN block of the ViT architecture with a mixture-of-experts block and compared it with the CNN and ResNet. Zhang et al. (2023) have tested their proposed method on 12 pre-trained models (CNN-, MLP-, and Transformer-based) in the context of test time domain adaptation. Except for ResNet, there are no comparative experiments on more than three different parameter sizes or pre-training data in the same model architecture, and no analysis of parameter sizes or datasets was conducted. Cha et al. (2022) and Arpit et al. (2022) conducted evaluation experiments using RegNetY-16GF as a pre-training model in addition to ResNet to evaluate the validity of their proposed method.

Yu et al. (2021) focused on only BiT, ViT, and ResNet architecture, Zhang et al. (2022) focused on only ViT, while Vishniakov et al. (2023) studied more in detail of ViT and ConvNeXt architecture in the context of downstream tasks. In addition, all related works mentioned above pay no attention to ECE under distributional shift.

**Trend of Larger Models and Datasets:** Recent evidence (Mikami et al., 2022) suggests that larger models and bigger pre-training datasets lead to better generalization in downstream tasks of Syn2Real (Yasarla et al., 2020). Not only in downstream tasks, this trend is also observed in the context of scaling laws (Kaplan et al., 2020; Zhai et al., 2022; Hestness et al., 2017). We systematically verify if this trend applies to OOD generalization and calibration scenarios.

On another note, Ruan et al. (2021) has taken on the challenge of improving OOD generalization performance in multimodal models using CLIP (Radford et al., 2021). However, as shown by Zhang et al. (2023). they point out that the images such as LAION-2B[*] that CLIP has trained on contain images analogous to the sketch domain of DomainNet (Peng et al., 2019), one of the evaluation datasets for OOD generalization. Since this could potentially undermine our problem setting for evaluating OOD generalization, although we have performed evaluation experiments on CLIP models as pre-trained models, we only present these settings and results in sections A.3 and D.7.

**Confidence Calibration Benchmark:** More recent studies have also evaluated modern architectures in comparison with calibration and robustness benchmarks, and Minderer et al. (2021) argue, as do Guo et al. (2017). that in independently and identically distributed (IID) settings, the ECE degrades with larger models. Conversely, in OOD scenarios, larger models tend to exhibit improved calibration performance (i.e., lower ECE) only under specific configurations, such as on certain models (e.g., BERT) and datasets (Tran et al., 2022; Kadavath et al., 2022; Dan & Roth, 2021). However, these findings have been limited to particular settings and lack generalizability. To address this, we conducted a far more comprehensive set of experiments, yielding results with broader applicability.

**Link between OOD Generalization and Calibration:** Recent theoretical study (Wald et al., 2021) has shown that multi-domain calibration is equivalent to invariant risk minimization (Arjovsky et al., 2019a), a popular OOD generalization technique. Some empirical observations indicate that techniques good at OOD generalization also achieve lower ECE (Ovadia et al., 2019; Immer et al., 2021; Naganuma & Kimura, 2023; Yoshida & Naganuma, 2024). We examine if this correlation holds for varying pre-trained models under distributional shift as well.

On another note, our experiments adopted pre-trained models from the PyTorch Image Models library (Wightman, 2019), also known as timm. It benchmarks models on ImageNet OOD test sets, such as ImageNet-

---

[*]https://laion.ai/blog/laion-5b/

Table 2: Datasets from the DomainBed (Gulrajani & Lopez-Paz, 2021) and WILDS (Koh et al., 2021) Banckmark used in our experiments. Domains with bold faces are used as out-of-distribution test domains, and others are used for (in-domain) training and validation.

| Dataset | Domains | #Images | #Classes |
|---|---|---|---|
| PACS | **art**, cartoon, photos, sketches | 9,991 | 7 |
| OfficeHome | **art**, clipart, product, real | 15,588 | 65 |
| DomainNet | clipart, infograph, painting, quickdraw, real, sketch | 586,575 | 345 |
| VLCS | **VOC2007**, Caltech101, LabelMe, SUN09 | 10,729 | 5 |
| Camelyon17 | **Hospital1**, Hospital2, Hospital3, Hospital4, Hospital5 | 455,954 | 2 |

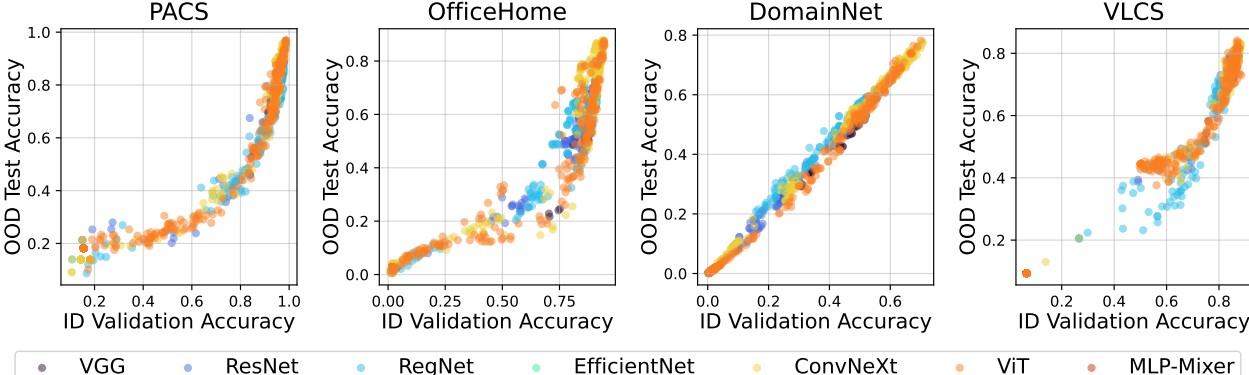

Figure 2: The relationship between in-domain (ID) validation accuracy and out-of-distribution (OOD) test accuracy of all experiments. When ID validation accuracy is high enough, ID and OOD accuracy are highly correlated, the "training-domain validation set" scheme can select near-optimal models.

R (Hendrycks et al., 2021a), ImageNet-A (Hendrycks et al., 2021b), ImageNet-v2 (Recht et al., 2019), and ImageNet-Real (Beyer et al., 2020) which is beneficial for our study.

## 4 Experimental Setup

Our experiments evaluate the OOD generalization accuracy (or error) of 100 pre-trained models fine-tuned on five OOD image classification datasets, namely PACS, OfficeHome, DomainNet, and VLCS, from the DomainBed benchmark (Gulrajani & Lopez-Paz, 2021) (see also table 2). In addition to accuracy, we use ECE as a complementary metric to assess the correctness of model confidence, which cannot be captured by accuracy (error) alone. In order to address concerns regarding ECE (Nixon et al., 2019), we also measured the ACE and confirmed the presence of identical trends, presented in section D.1. Additionally, we conducted medical image classification experiments using Camelyon17 datasets from WILDS benchmark (Koh et al., 2021) to cover other types of OOD generalization task evaluation (section D.6).

Pre-trained models are selected from the `timm` library (Wightman, 2019), which offers a vast collection of over a thousand pre-trained image classification models. Among them, we use models pre-trained on families of ImageNet to minimize data overlap with the DomainBed and WILDS datasets, since our interest lies in OOD generalization. Specifically, we adopt pre-training datasets of ImageNet-1k (1k categories, 1.4M images), ImageNet-12k (12k categories, 12M images), ImageNet-21k (21k categories, 14M images), and ImageNet-22k (22k categories, 14M images). While `timm` also offers models pre-trained on larger datasets, such as LAION (Schuhmann et al., 2022) consisting of billions of images, we refrain from utilizing them due to their excessive size, which raises concerns about the potential inclusion of OOD test data. In addition to the models pre-trained on image data only, we validate CLIP models, which are pre-trained on image-text pairs (section A.3). Selected 100 pre-trained models, varying parameter sizes from 3M to 600M, are listed in table A.1. Further details are explained in section A.1.

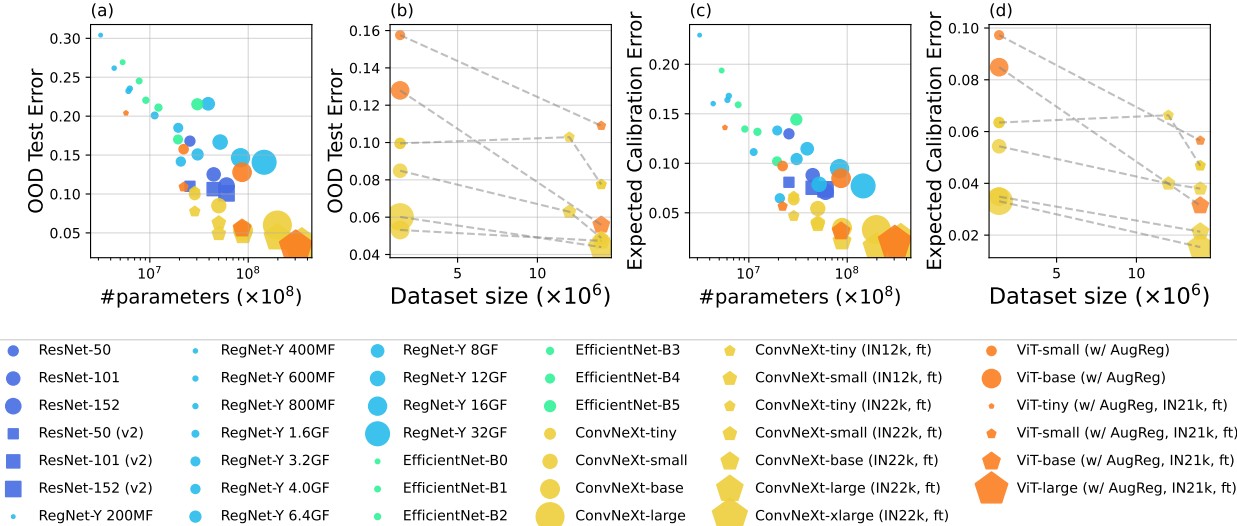

Figure 3: (a,b) OOD test error rates (lower is better) with respect to the number of parameters (a, log-scale) and the pre-training dataset sizes (b) on the PACS dataset. In the right panel, the same model architectures with different pre-training datasets are connected by dashed lines. We can observe that **the number of parameters and the pre-training dataset sizes contribute to the OOD generalization**. Presented models are selected for a better view. Results of other OOD datasets with all models are presented in figs. D.5 to D.7. (c,d) OOD expected calibration error (lower is better) with respect to the number of parameters (c) and the pre-training dataset sizes (d) on the PACS dataset. In the right panel, the same model architectures with different pre-training datasets are connected by dashed lines. We can see trends that **the number of parameters and the pre-training dataset sizes improve ECE**. Presented models are selected for the better view. Results of other OOD datasets with all models are presented in figs. D.5 to D.7.

For fine-tuning, we follow DomainBed's experimental protocol. The key difference from the original protocol is that we use various pre-trained models from `timm`, rather than fixing the pre-trained model only to `ResNet-50`. To prioritize the exploration of models under fixed computational constraints, we adopt a simple evaluation approach different from the original DomainBed. Namely, we select each dataset's first domain listed in table 2 as the OOD test domain and use the remaining domains as (in-domain) training and validation data.

Another difference from DomainBed is the choice of optimizer for fine-tuning. We adopt SGD with a momentum of 0.9 as a default optimizer rather than Adam (Kingma & Ba, 2015), as SGD with Momentum is more effective in OOD generalization (Naganuma et al., 2023) (we confirm this in figs. C.2 and C.3). In addition, we fine-tune models with adaptive optimizers when they are used for pre-training: RMSProp for `EfficientNets`, AdamW for `ConvNeXts`, and Adam for `ViTs`.

Their hyperparameters, learning rates, and weight decay rates are selected using in-domain validation data, as we cannot access OOD data. This selection is referred to as the "training-domain validation set" scheme in the DomainBed benchmark (Gulrajani & Lopez-Paz, 2021) and is capable of finding near-optimal models when in-domain validation accuracy is high enough (fig. 2). Further details on hyperparameter selection are described in section A.2. We also run experiments over three different random seeds, and their effects on the final results are negligible, as shown in table C.1.

## 5   Results

In the remaining text, we report OOD generalization performance and ECE on the held-out test domain data of models selected by the above-mentioned validation scheme. Full results and a detailed analysis for each model architecture are shown in section D.4 and section D.5.

### 5.1 Scaling Trend: OOD Generalization

Our comprehensive experiments reveal several consistent and significant trends in how pre-trained model selection impacts out-of-distribution generalization performance.

First, we find that increasing the number of parameters in pre-trained models leads to noticeable gains in OOD accuracy across almost all model architectures examined, as shown in fig. 3. Similarly, pre-trained models on larger dataset sizes yield substantial OOD performance improvements. These trends hold not just for the PACS examples shown here but also generalize to other evaluation datasets for domain generalization (figs. D.6 and D.7). Second, when comparing models with similar parameter sizes, we observe that ViT and ConvNeXt architectures tend to achieve higher OOD accuracy than conventional CNN architectures like ResNet and EfficientNet. This suggests the benefits of more recent model architectures specialized for computer vision tasks.

As a remarkable observation, we also find that OOD error decreases almost linearly on a log-scale of parameters as the pre-trained model size increases as shown in fig. 3 (a). This implies the **sign of scaling laws** and hints at shared underlying mechanics that drive performance gains in downstream domain generalization tasks. Lastly, through targeted experiments on ConvNeXt and ViT models, we verify that increasing the pre-training dataset size significantly boosts OOD accuracy. This further underscores the importance of large-scale pre-training data for enhancing OOD generalization.

In summary, our extensive empirical results reveal consistent trends that bigger pre-trained model sizes and larger pre-training datasets lead to substantial and reliable improvements in out-of-distribution generalization performance across diverse model architectures and evaluation datasets. These insights provide practical guidance for pre-trained model selection when targeting OOD tasks.

### 5.2 A Deeper Analysis from the Perspective of Confidence Calibration

To uncover the mechanisms behind the scaling trends in OOD generalization observed in fine-tuning pre-trained models, we evaluate confidence calibration using ECE. Calibration is adopted for two main reasons: (1) Calibration in multi-domain settings is theoretically equivalent to invariant feature learning for OOD generalization (Wald et al., 2021), and it has been empirically shown to improve OOD accuracy and invariant feature learning (Ovadia et al., 2019; Immer et al., 2021; Naganuma & Kimura, 2023; Yoshida & Naganuma, 2024), whereas accuracy or error alone cannot capture the degree of invariant feature learning; (2) Other promising generalization metrics, such as curvature, are challenging to compare across model architectures, whereas calibration, based on output distributions, is more suitable for our evaluation. The results are shown in fig. 3 (c, d).

First, we find that increasing the number of parameters in pre-trained models leads to lower ECE, mirroring the trends observed for OOD generalization. Using pre-trained models trained on larger dataset sizes also improves calibration. From these results, we infer that increasing model size or the amount of training data during pre-training helps to mitigate domain-specific overconfidence in downstream tasks, which in turn promotes OOD generalization, including through mechanisms like invariant feature learning.

Furthermore, these results contrast with prior work in IID settings, found modern deep networks like ResNet-110 to have significantly higher ECE than small networks like LeNet-5, despite their higher accuracy (Guo et al., 2017; Minderer et al., 2021). In line with Minderer et al. (2021), we demonstrate that in OOD settings, at even larger parameter scales beyond ResNet-110, the trend reverses, and ECE starts to decrease with model parameter size. Notably, ECE reduces almost linearly as pre-trained model size increases on a log scale, a novel finding that parallels the scaling laws for OOD error. This strengthens the claim that, in OOD settings, increasing model size significantly supports improved calibration.

In summary, we confirm that increasing the size of the pre-trained model and pre-training data boosts OOD generalization. Additionally, a deeper investigation from the perspective of calibration reveals that a similar trend holds for ECE, showing that larger models and datasets during pre-training help mitigate overconfidence in downstream tasks. Our results also overturn prior observations in IID settings that modern deep networks

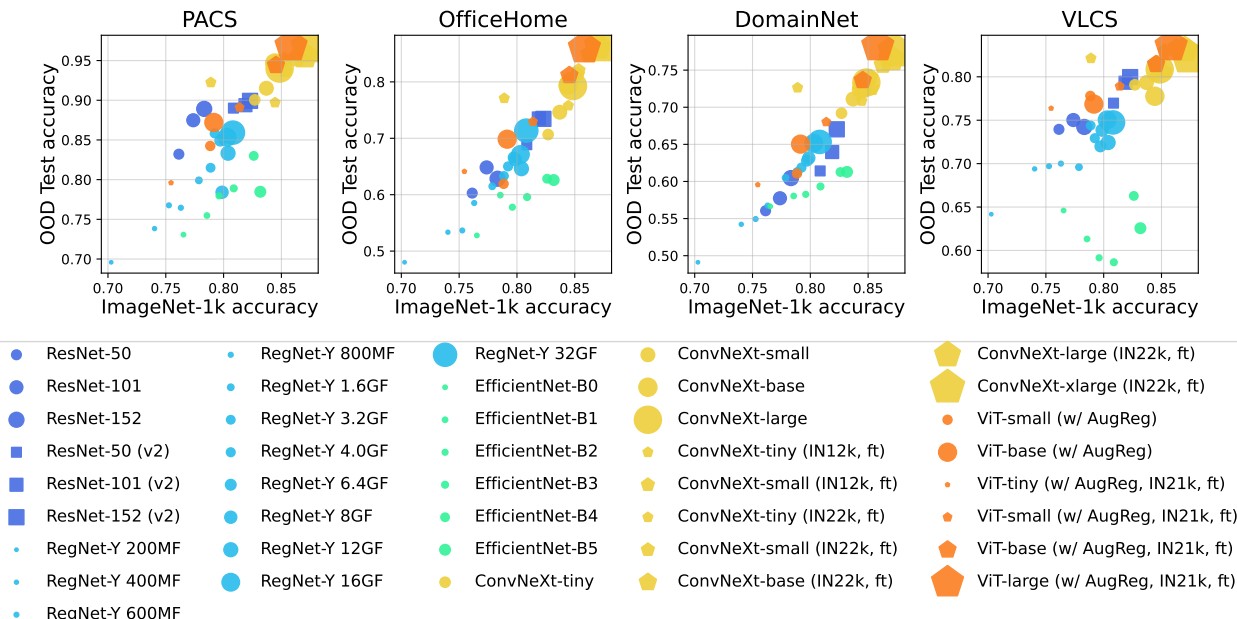

Figure 4: Correlation between ImageNet-1k validation accuracy and out-of-distribution test accuracy. OOD test accuracy shows a strong positive correlation with ImageNet-1k validation accuracy. However, EfficientNets fall below the trend of other models, suggesting that *they may be overfitting to ImageNet-1k.*

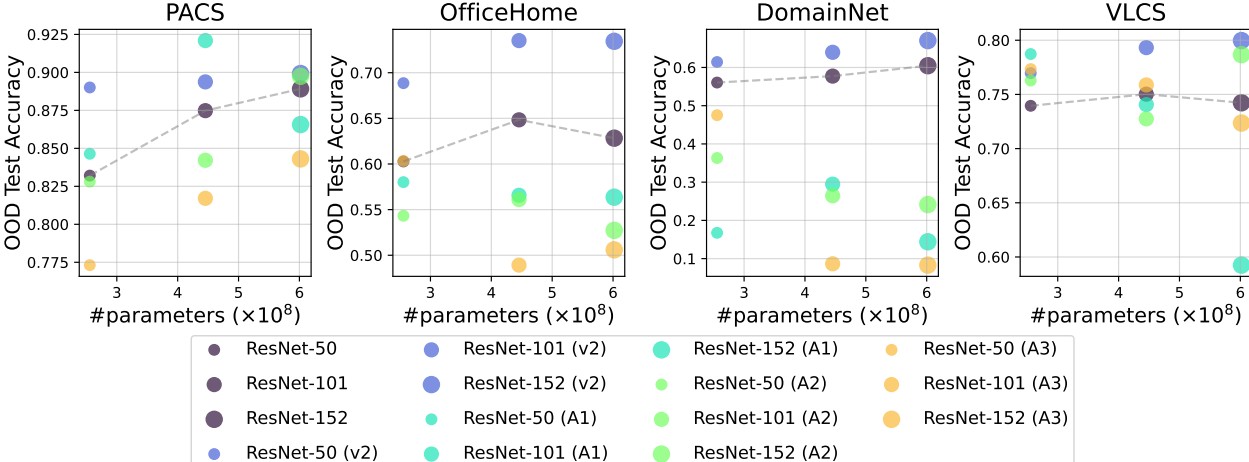

Figure 5: Out-of-distribution test accuracy of ResNets (He et al., 2016) with different training schemes. The results of the original ResNets are connected by dashed lines. Although extensive training recipes (Wightman et al., 2021) improve ImageNet-1k performance, their fine-tuned OOD results often underperform the original ones.

suffer from higher ECE, underscoring the potential of large-scale pre-trained models to address calibration challenges specifically in OOD contexts.

## 5.3 Pre-trained Models can Overfit to Training Set

We observe that some pre-trained models are likely to overfit to training set such as ImageNet-1k. Figure 4 presents the relationship between ImageNet-1k validation accuracy and OOD test accuracy. As can be seen, the EfficientNet models tend to fall below the trendlines of other models, suggesting that they are more prone to overfitting to ImageNet-1k and do not transfer well to OOD datasets. This overfit could be attributed to EfficientNets being designed by neural architecture search (Tan & Le, 2019), where the components of networks

are searched to maximize validation performance. Although improvement of pre-training methods, such as using extensive data augmentation (Cubuk et al., 2019; Xie et al., 2020a), enhances OOD generalization performance (fig. D.11), overfit trends unchanged as evident in fig. D.8. This figure also depicts that fine-tuned EfficientNets degenerates calibration on the OOD datasets. In contrast, RegNets (Radosavovic et al., 2020) seems to avoid overfitting, probably because they are optimized at a meta-level (design principles), not at a component level.

Wightman et al. (2021) reported that ResNet-50 with renewed training recipes, including additional data augmentation and longer training, attains more than 5% ImageNet-1k accuracy improvement over the original setting. Figure 5 displays the comparison of the ResNets pre-trained with the original and the updated recipes: a new training recipe introduced in torchvision* referred to as v2 and others proposed in (Wightman et al., 2021) referred to as A1-A3. Although models trained with these new training recipes improve ImageNet-1k performance, A1-A3 often perform inferior to the original ones under the same fine-tuning setting. Contrarily, the v2 models consistently outperform the original models. These results may indicate that certain aspects of pre-training strategies in Wightman et al. (2021) may lead to overfitting to ImageNet-1k. Disentangling the specific factors causing the OOD performance degeneration is challenging, due to the various techniques employed in these recipes. However, it is possible to guess that binary cross-entropy loss and/or Adam optimizer during pre-training are attributed to the issue. It is also worth noting that extensive data augmentation, such as RandAugment (Cubuk et al., 2020) and AugMix (Hendrycks* et al., 2020), improves the OOD generalization as in fig. D.14.

These overfit phenomena of EfficientNets and enhanced ResNets cannot be observed on ImageNet OOD test sets, such as ImageNet-A, as presented in fig. D.10. Rather, EfficientNets and ResNets with A1-A3 recipes are aligned with the correlations between the performance of ImageNet-1k and ImageNet OOD test sets. These findings suggest that the generalization on pre-training datasets, such as ImageNet-1k, does not directly translate to the generalization after fine-tuning.

### 5.4   Other Findings

Above three are the major findings, but other findings of this study are briefly described. Details are given in section B. First, in the pre-training phase, supervised pre-training with AugReg as a data augmentation approach improves OOD test accuracy compared to self-supervised methods like DINO and MAE. Second, as prior work (Naganuma et al., 2023) suggests advantages for SGD with momentum over adaptive optimizers for OOD generalization, this study confirmed SGD with momentum consistently outperforms other optimizers when fine-tuning diverse architectures, including ViT and EfficientNet. Third, the impact of weight decay on OOD performance varies across datasets. While stronger regularization improves calibration, it can hinder OOD accuracy on datasets with visually distinct domains, as shown in fig. B.2. The ablation studies for sensitivity and seed of hyperparameters are shown in section C.

Analysis of ImageNet variant datasets for robustness benchmarks suggests a positive correlation between model robustness on ImageNet variants and domain generalization performance, with improvements on sampling bias datasets showing a stronger correlation compared to robustness benchmarks as shown in D.9.

Additionally, section D.6 shows the results evaluated on the Camelyon17 datasets, aligning with the trends observed in DomainBed. Section D.7 shows the results of CLIP models, which align with other models in terms of test accuracy.

## 6   Discussion and Conclusion

In this work, we demonstrated the signs of scaling laws for OOD generalization and calibration of pre-trained models in the image classification setting. This conclusion is drawn from a comprehensive study involving five datasets and over 100 different combinations of pre-training model architectures, pre-training datasets, model sizes, training strategies, and hyperparameter choices.

---

*Details are described in https://pytorch.org/blog/how-to-train-state-of-the-art-models-using-torchvision-latest-primitives.

Our results are also useful to computer vision practitioners due to their prescriptive nature: our systematic study on over 100 pre-trained visual models yielded that in the OOD setting with pretraining, larger architectures using larger training sets not only achieve better out-of-distribution generalization but also improve calibration, further reducing overconfidence in downstream tasks. We also found that non-generic training strategies—for example, the ones specifically designed to maximize in-distribution performance for ImageNet-1k on smaller models—do not achieve good OOD performance, indicating overfitting to the pre-training dataset.

In light of these results, we summarize our findings as the following guidelines on pre-trained model selection for computer vision practitioners:

1. Use large pre-trained models,

2. Use models pre-trained on large datasets,

3. Use models pre-trained using generic methodology (i.e. not specifically designed to maximize in-distribution performace on a specific architecture).

Following these guidelines can substantially improve OOD generalization and calibration capabilities.

Our results can also prove to be useful to researchers, as they provide guidance for future work in the area. In particular, one important by-product of our results is a newfound appreciation of model selection as a strong baseline. This point is important because recent work providing new, complex methodology, yields results that are inferior to the ones we provide here based on simple model selection. Future work on computation-efficient methods for the OOD setting is critical; we hope that our results will help researchers build those new, efficient methods on top of solid baselines.

## Acknowledgments

Our deepest gratitude goes out to the anonymous reviewers and the Action Editor, whose invaluable insights substantially enhanced the quality of this manuscript. The computational resources instrumental to this study were provided under the auspices of the "ABCI Grand Challenge" Program, National Institute of Advanced Industrial Science and Technology.

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

# Appendix Table of Contents

## A  Detailed Experimental Settings

All experiments' codes are modifications provided by the authors who introduced the datasets (Gulrajani & Lopez-Paz, 2021; Koh et al., 2021). Licenses of the codes are MIT license for DomainBed (Gulrajani & Lopez-Paz, 2021) and WILDS (Koh et al., 2021). The license of the pre-trained model used in the experiment is Apache 2.0 license [*].

We performed our experiment with ABCI Cluster. For ABCI Cluster, each node is composed of NVIDIA Tesla V100×4GPU and Intel Xeon Gold 6148 2.4 GHz, 20 Cores×2CPU. As a software environment, we use Red Hat 4.8.5, gcc 7.4, Python 3.6.5, Pytorch 1.6.0, cuDNN 7.6.2, and CUDA 10.0. The total amount of computation is over 120,000 GPU hours.

Please see the code here: `https://github.com/Hiroki11x/Timm_OOD_Calibration`

### A.1  Models used in the Experiments

Table A.1 present all models, in a total of 100, we used in the experiments, sorted by ImageNet-1k accuracy.

Table A.1: Models from `timm` used in the experiments with ImageNet-1k validation accuracy and the number of parameters sorted by ImageNet-1k validation accuracy. "ft" stands for finetuning on ImageNet-1k. Self-supervised pre-trained models present "NaN" accuracy.

| Model | IN-1k Acc. | # Param. ($\times 10^6$) |
|---|---|---|
| ConvNeXt-xlarge (IN22k, ft) (Liu et al., 2022) | 87.33 | 350.20 |
| | *Continued on next page* | |

---

[*] `https://github.com/huggingface/pytorch-image-models`

Table A.1: Models from `timm` used in the experiments with ImageNet-1k validation accuracy and the number of parameters sorted by ImageNet-1k validation accuracy. "ft" stands for finetuning on ImageNet-1k. Self-supervised pre-trained models present "NaN" accuracy.

| Model | IN-1k Acc. | # Param. ($\times 10^6$) |
|---|---|---|
| ConvNeXt-large (IN22k, ft) (Liu et al., 2022) | 87.03 | 197.77 |
| ConvNeXt-base (IN22k, ft) (Liu et al., 2022) | 86.27 | 88.59 |
| EfficientNet-B5 (w/ Noisy Student, JFT, ft) (Tan & Le, 2019; Xie et al., 2020b) | 86.09 | 30.39 |
| ViT-large (w/ AugReg, IN21k, ft) (Dosovitskiy et al., 2020; Steiner et al., 2022) | 85.83 | 304.33 |
| ConvNeXt-small (IN12k, ft) (Liu et al., 2022) | 85.33 | 50.22 |
| ConvNeXt-small (IN22k, ft) (Liu et al., 2022) | 85.26 | 50.22 |
| EfficientNet-B4 (w/ Noisy Student, JFT, ft) (Tan & Le, 2019; Xie et al., 2020b) | 85.16 | 19.34 |
| ConvNeXt-large (Liu et al., 2022) | 84.85 | 197.77 |
| ViT-base (w/ AugReg, IN21k, ft) (Dosovitskiy et al., 2020; Steiner et al., 2022) | 84.53 | 86.57 |
| ConvNeXt-tiny (IN12k, ft) (Liu et al., 2022) | 84.45 | 28.59 |
| ConvNeXt-base (Liu et al., 2022) | 84.43 | 88.59 |
| EfficientNet-B5 (w/ AdvProp) (Tan & Le, 2019; Xie et al., 2020a) | 84.26 | 30.39 |
| EfficientNet-B3 (w/ Noisy Student, JFT, ft) (Tan & Le, 2019; Xie et al., 2020b) | 84.06 | 12.23 |
| EfficientNet-B5 (w/ RandAugment) (Tan & Le, 2019; Cubuk et al., 2020) | 83.81 | 30.39 |
| ConvNeXt-small (Liu et al., 2022) | 83.70 | 50.22 |
| EfficientNet-B5 (w/ AutoAugment) (Tan & Le, 2019; Cubuk et al., 2019) | 83.69 | 30.39 |
| ResNet-152 (A1H) (Wightman et al., 2021) | 83.45 | 60.19 |
| EfficientNet-B4 (w/ AdvProp) (Tan & Le, 2019; Xie et al., 2020a) | 83.25 | 19.34 |
| EfficientNet-B5 (Tan & Le, 2019) | 83.18 | 30.39 |
| EfficientNet-B4 (w/ AutoAugment) (Tan & Le, 2019; Cubuk et al., 2019) | 83.02 | 19.34 |
| ResNet-101 (A1H) (Wightman et al., 2021) | 82.78 | 44.55 |
| ResNet-152 (A1) (Wightman et al., 2021) | 82.73 | 60.19 |
| ConvNeXt-tiny (Liu et al., 2022) | 82.70 | 28.59 |
| EfficientNet-B4 (Tan & Le, 2019) | 82.61 | 19.34 |
| ResNet-152 (A2) (Wightman et al., 2021) | 82.61 | 60.19 |
| EfficientNet-B2 (w/ Noisy Student, JFT, ft) (Tan & Le, 2019; Xie et al., 2020b) | 82.38 | 9.11 |
| ResNet-101 (A1) (Wightman et al., 2021) | 82.32 | 44.55 |
| ResNet-152 (v2) (He et al., 2016) | 82.29 | 60.19 |
| ResNet-101 (A2) (Wightman et al., 2021) | 82.24 | 44.55 |
| ResNet-101 (v2) (He et al., 2016) | 81.89 | 44.55 |
| EfficientNet-B3 (w/ AdvProp) (Tan & Le, 2019; Xie et al., 2020a) | 81.83 | 12.23 |
| ViT-base (IN21k, ft) (Dosovitskiy et al., 2020) | 81.79 | 86.57 |
| EfficientNet-B3 (w/ AutoAugment) (Tan & Le, 2019; Cubuk et al., 2019) | 81.64 | 12.23 |
| EfficientNet-lite4 (Tan & Le, 2019) | 81.53 | 13.01 |
| EfficientNet-B1 (w/ Noisy Student, JFT, ft) (Tan & Le, 2019; Xie et al., 2020b) | 81.39 | 7.79 |
| ViT-small (w/ AugReg, IN21k, ft) (Dosovitskiy et al., 2020; Steiner et al., 2022) | 81.38 | 22.05 |
| ResNet-50 (A1) (Wightman et al., 2021) | 81.21 | 25.56 |
| ResNet-50 (D) (Wightman et al., 2021) | 80.97 | 25.56 |
| ResNet-50 (C1) (Wightman et al., 2021) | 80.91 | 25.56 |
| EfficientNet-B3 (Tan & Le, 2019) | 80.88 | 12.23 |
| ResNet-50 (C2) (Wightman et al., 2021) | 80.87 | 25.56 |
| ResNet-50 (v2) (He et al., 2016) | 80.85 | 25.56 |
| RegNet-Y 32GF (Radosavovic et al., 2020) | 80.81 | 145.05 |
| ResNet-50 (A2) (Wightman et al., 2021) | 80.77 | 25.56 |
| ResNet-50 (B1K) (Wightman et al., 2021) | 80.71 | 25.56 |
| ResNet-50 (A1H) (Wightman et al., 2021) | 80.68 | 25.56 |
| ResNet-152 (A3) (Wightman et al., 2021) | 80.55 | 60.19 |

Table A.1: Models from `timm` used in the experiments with ImageNet-1k validation accuracy and the number of parameters sorted by ImageNet-1k validation accuracy. "ft" stands for finetuning on ImageNet-1k. Self-supervised pre-trained models present "NaN" accuracy.

| Model | IN-1k Acc. | # Param. $(\times 10^6)$ |
|---|---|---|
| ResNet-50 (B2K) (Wightman et al., 2021) | 80.45 | 25.56 |
| RegNet-Y 12GF (Radosavovic et al., 2020) | 80.39 | 51.82 |
| EfficientNet-B2 (w/ AdvProp) (Tan & Le, 2019; Xie et al., 2020a) | 80.31 | 9.11 |
| RegNet-Y 16GF (Radosavovic et al., 2020) | 80.30 | 83.59 |
| ViT-base (SAM) (Dosovitskiy et al., 2020; Foret et al., 2021) | 80.24 | 86.57 |
| EfficientNet-B2 (w/ AutoAugment) (Tan & Le, 2019; Cubuk et al., 2019) | 80.08 | 9.11 |
| ResNet-50 (AugMix) (He et al., 2016; Hendrycks* et al., 2020) | 79.98 | 25.56 |
| RegNet-Y 8GF (Radosavovic et al., 2020) | 79.87 | 39.18 |
| ResNet-50 (RA) (He et al., 2016; Cubuk et al., 2020) | 79.84 | 25.56 |
| ResNet-101 (A3) (Wightman et al., 2021) | 79.81 | 44.55 |
| EfficientNet-lite3 (Tan & Le, 2019) | 79.81 | 8.20 |
| RegNet-Y 6.4GF (Radosavovic et al., 2020) | 79.71 | 30.58 |
| EfficientNet-B2 (Tan & Le, 2019) | 79.60 | 9.11 |
| EfficientNet-B1 (w/ AdvProp) (Tan & Le, 2019; Xie et al., 2020a) | 79.28 | 7.79 |
| RegNet-Y 4.0GF (Radosavovic et al., 2020) | 79.23 | 20.65 |
| ViT-base (w/ AugReg) (Dosovitskiy et al., 2020; Steiner et al., 2022) | 79.15 | 86.57 |
| ConvNeXt-tiny (IN22k, ft) (Liu et al., 2022) | 78.90 | 28.59 |
| RegNet-Y 3.2GF (Radosavovic et al., 2020) | 78.88 | 19.44 |
| ViT-small (w/ AugReg) (Dosovitskiy et al., 2020; Steiner et al., 2022) | 78.85 | 22.05 |
| EfficientNet-B1 (w/ AutoAugment) (Tan & Le, 2019; Cubuk et al., 2019) | 78.83 | 7.79 |
| EfficientNet-B0 (w/ Noisy Student, JFT, ft) (Tan & Le, 2019; Xie et al., 2020b) | 78.67 | 5.29 |
| EfficientNet-B1 (Tan & Le, 2019) | 78.56 | 7.79 |
| ResNet-152 (He et al., 2016) | 78.32 | 60.19 |
| ResNet-50 (A3) (Wightman et al., 2021) | 78.05 | 25.56 |
| RegNet-Y 1.6GF (Radosavovic et al., 2020) | 77.86 | 11.20 |
| EfficientNet-lite2 (Tan & Le, 2019) | 77.46 | 6.09 |
| ResNet-101 (He et al., 2016) | 77.38 | 44.55 |
| EfficientNet-B0 (w/ AdvProp) (Tan & Le, 2019; Xie et al., 2020a) | 77.09 | 5.29 |
| EfficientNet-B0 (w/ AutoAugment) (Tan & Le, 2019; Cubuk et al., 2019) | 76.83 | 5.29 |
| EfficientNet-lite1 (Tan & Le, 2019) | 76.64 | 5.42 |
| MLP-MixereB (IN21k, ft) (Tolstikhin et al., 2021) | 76.60 | 59.88 |
| EfficientNet-B0 (Tan & Le, 2019) | 76.53 | 5.29 |
| RegNet-Y 800MF (Radosavovic et al., 2020) | 76.30 | 6.26 |
| ResNet-50 (He et al., 2016) | 76.13 | 25.56 |
| ViT-tiny (w/ AugReg, IN21k, ft) (Dosovitskiy et al., 2020; Steiner et al., 2022) | 75.45 | 5.72 |
| RegNet-Y 600MF (Radosavovic et al., 2020) | 75.27 | 6.06 |
| EfficientNet-lite0 (Tan & Le, 2019) | 74.83 | 4.65 |
| RegNet-Y 400MF (Radosavovic et al., 2020) | 74.03 | 4.34 |
| VGG19 (Simonyan & Zisserman, 2015) | 72.38 | 143.67 |
| MLP-Mixer-L (IN21k, ft) (Tolstikhin et al., 2021) | 72.05 | 208.20 |
| VGG16 (Simonyan & Zisserman, 2015) | 71.59 | 138.36 |
| RegNet-Y 200MF (Radosavovic et al., 2020) | 70.28 | 3.16 |
| VGG13 (Simonyan & Zisserman, 2015) | 69.93 | 133.05 |
| VGG11 (Simonyan & Zisserman, 2015) | 69.02 | 132.86 |
| ViT-base (patch 16, MAE) (Dosovitskiy et al., 2020; He et al., 2022) | NaN | 85.80 |
| ViT-huge (patch 14, MAE) (Dosovitskiy et al., 2020; He et al., 2022) | NaN | 630.80 |
| ViT-large (patch 14, MAE) (Dosovitskiy et al., 2020; He et al., 2022) | NaN | 303.30 |

Table A.1: Models from `timm` used in the experiments with ImageNet-1k validation accuracy and the number of parameters sorted by ImageNet-1k validation accuracy. "ft" stands for finetuning on ImageNet-1k. Self-supervised pre-trained models present "NaN" accuracy.

| Model | IN-1k Acc. | # Param. ($\times 10^6$) |
|---|---|---|
| ViT-small (DINO) (Dosovitskiy et al., 2020; Caron et al., 2021) | NaN | 21.70 |
| ViT-base (DINO) (Dosovitskiy et al., 2020; Caron et al., 2021) | NaN | 85.80 |
| CLIP (ResNet-50, Zero-shot) (Radford et al., 2021) | 59.6[*] | 38.31 |
| CLIP (ResNet-101, Zero-shot) (Radford et al., 2021) | 62.2[*] | 56.26 |
| CLIP (ViT-B/32, Zero-shot) (Radford et al., 2021) | 63.2[*] | 87.85 |

## A.2  Model Selection and Hyperparameter Search

We adopted the most common model selection scheme, "training-domain validation set" of the DomainBed paper (Gulrajani & Lopez-Paz, 2021). We adopted grid search over learning rates from $\{1.0 \times 10^{-4}, 5.0 \times 10^{-4}, 1.0 \times 10^{-3}, 5.0 \times 10^{-3}, 1.0 \times 10^{-2}, 5.0 \times 10^{-2}, 1.0 \times 10^{-1}, 5.0 \times 10^{-1}\}$ and weight-decay rates from $\{1.0 \times 10^{-4}, 1.0 \times 10^{-3}, 1.0 \times 10^{-2}\}$ of SGD with a momentum rate of 0.9, because Momentum SGD is known to be more effective in OOD environment (Naganuma et al., 2023). Additionally, for models trained with adaptive optimizers, such as Adam (Kingma & Ba, 2015) and AdamW (Loshchilov & Hutter, 2017), we also tried the same optimizers and searched learning rates from $\{1.0 \times 10^{-4}, 5.0 \times 10^{-4}, 1.0 \times 10^{-3}, 5.0 \times 10^{-3}, 1.0 \times 10^{-2}, 5.0 \times 10^{-2}, 1.0 \times 10^{-1}, 5.0 \times 10^{-1}\}$. The batch size and the number of training iterations were not subjects of our exploration: the batch size was fixed to 32, and the number of training iterations was fixed to 5,000. Figures C.2 and C.3 present the effect of hyperparameter selection on in-domain validation accuracy, which was used to select hyperparameters, and out-of-distribution test accuracy, which was used to report the final performance.

## A.3  Experimental Protocol of CLIP Model

Our research deliberately focused on fine-tuning pre-trained models, rather than zero-shot learning with large-scale models like CLIP (Radford et al., 2021). While we recognize the novelty and significance of zero-shot learning, we prioritized maintaining a specific research scope, especially as fine-tuning remains necessary for domain-specific models, even with advancements in Large Language Models and Large Vision Models, as shown in (Wei et al., 2023). Although it is questionable whether the CLIP pre-trained model can quantify true domain generalization in terms of pre-training on large datasets that contain images corresponding to such domains as the sketch domain, we evaluate the CLIP model on four different datasets from DomainBed.

We conducted the experiments according to Ruan et al.. In particular, and evaluated three different vision encoders for CLIP: ResNet-50[*], ResNet-101[*], and ViT-B/32 [*].

For zero-shot inference, as shown in figure 1 (3) of the paper by Radford et al., input the data to be inferred in the image encoder and "a photo of a {class}" in the text encoder.

## A.4  Datasets

Among the datasets within the DomainBed (Gulrajani & Lopez-Paz, 2021) and WILDS benchmarks (Koh et al., 2021), we selected specific datasets detailed in table A.2 for our experiments. The initial domain

---

[*]We use IN-1K accuracy reported in Radford et al. (2021)

[*]https://openaipublic.azureedge.net/clip/models/afeb0e10f9e5a86da6080e35cf09123aca3b358a0c3e3b6c78a7b63bc04b6762/RN50.pt

[*]https://openaipublic.azureedge.net/clip/models/8fa8567bab74a42d41c5915025a8e4538c3bdbe8804a470a72f30b0d94fab599/RN101.pt

[*]https://openaipublic.azureedge.net/clip/models/40d365715913c9da98579312b702a82c18be219cc2a73407c4526f58eba950af/ViT-B-32.pt

outlined in table A.2 served as the test domain for each dataset, while the remaining domains were employed for training and validation.

Within DomainBed (Gulrajani & Lopez-Paz, 2021), the VLCS dataset (Fang et al., 2013) compiles natural images from various image datasets, which are treated as distinct domains. For example, in the PACS dataset (Li et al., 2017), a label such as "dog" is associated with several domains including photographs, sketches, art, and cartoons, which are used for domain generalization tasks. OfficeHome (Venkateswara et al., 2017) and DomainNet (Peng et al., 2019), similarly, possess domains that are analogous in nature; however, OfficeHome focuses on objects typically found in offices like chairs, whereas DomainNet includes a wider array of objects.

To evaluate task trends beyond visual domain generalization, we examined Camelyon17 (Bandi et al., 2018) from the WILDS benchmark as a medical imaging dataset. Camelyon17 includes histopathological images sourced from five different hospitals, each treated as a separate domain.

| Dataset | Domains | #Exampls | #Categories |
|---|---|---|---|
| PACS | `art`, `cartoon`, `photos`, `sketches` | 9,991 | 7 |
| OfficeHome | `art`, `clipart`, `product`, `real` | 15,588 | 65 |
| DomainNet | `clipart`, `infograph`, `painting`, `quickdraw`, `real`, `sketch` | 586,575 | 345 |
| VLCS | `VOC2007`, `Caltech101`, `LabelMe`, `SUN09`, | 10,729 | 5 |
| Camelyon17 | `Hospital1`, `Hospital2`, `Hospital3`, `Hospital4`, `Hospital5` | 455,954 | 2 |

Table A.2: Datasets from the DomainBed and WILDS benchmark used in the experiments.

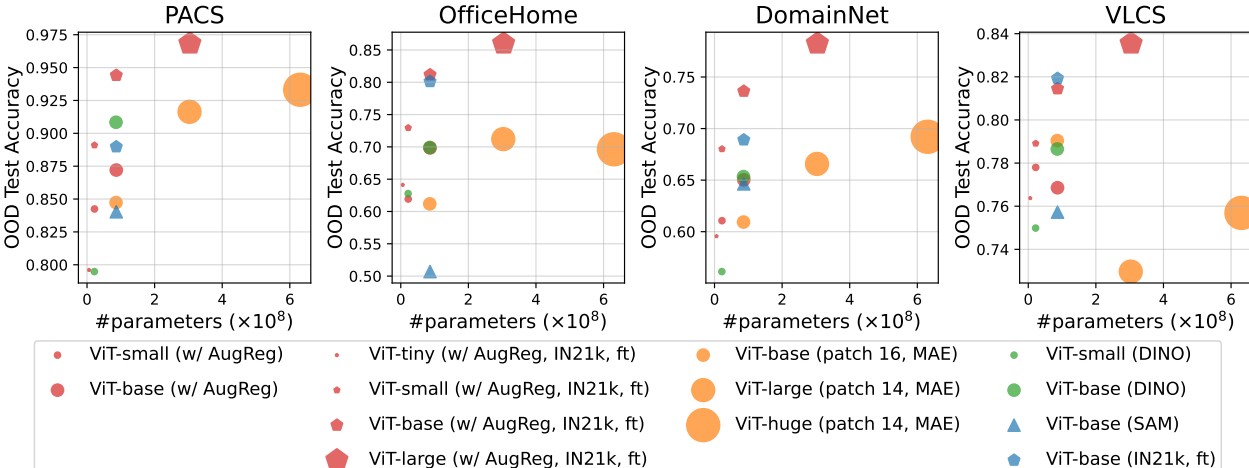

Figure B.1: Out-of-distribution test accuracy of ViT (Dosovitskiy et al., 2020) with various training schemes. Comparing the same architectures aligned in vertical lines, supervised pre-training outperforms self-supervised pre-training (DINO (Caron et al., 2021) and MAE (He et al., 2022)). Also interestingly, pre-training with the SAM optimizer (Foret et al., 2021) yields inferior performance to its SGD counterpart.

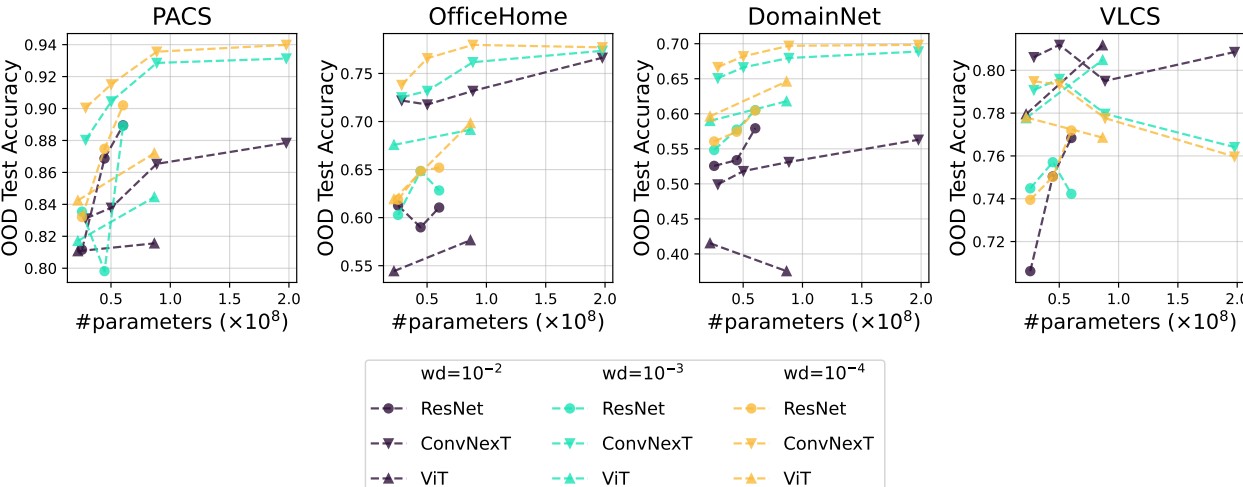

Figure B.2: OOD test accuracy with respect to the number of model parameters with different weight decay rates of momentum SGD. For the datasets different from ImageNet, PACS, OfficeHome, and DomainNet, smaller weight-decay rates result in better performance. On the other hand, for datasets similar to ImageNet and VLCS, larger weight-decay rates improve performance.

# B    Other Findings

This section describes other findings of the factors in the fine-tuning and pre-training phase and correlation behavior of datasets.

## B.1    Factors in Pre-training Phase

Through focused analysis, we gained several insights into optimal pre-training strategies for ViT architectures. Our key findings are as follows.

For supervised learning, pre-training with data augmentation and regularization through AugReg (Steiner et al., 2022) improves OOD test accuracy. Examining self-supervised training methods, we find models pre-trained with DINO and MAE perform worse than supervised learning on OOD test accuracy. This result that using self-supervised learning in the pre-training phase degrades performance in the downstream task is

in agreement with the results of (Goldblum et al., 2023). More critically, pre-trained with the optimizer of Sharpness Aware Minimization (Foret et al., 2021) exhibit overfitting on ImageNet-1k that degrades OOD generalization, as evidenced by their lower OOD versus in-distribution accuracy (fig. D.15).

## B.2   Factors in Fine-tuning Phase

**Optimizer Selection:** We also performed comprehensive experiments to evaluate optimizer selections for fine-tuning pre-trained models. Prior work by Naganuma et al. (2023) showed SGD with momentum outperforming adaptive optimizers like Adam for OOD generalization across benchmarks, including DomainBed. However, their experiments on DomainBed were limited to ResNet-50 models, leaving the question of optimal optimizers for diverse model architectures open.

While our main experiments use SGD with momentum, different architectures like EfficientNet and ViT were originally proposed with other optimizers for pre-training. We directly compare fine-tuning these models with their original pre-training optimizers versus SGD with momentum. Our results in fig. C.3 show SGD with momentum achieves superior OOD test accuracy in nearly all cases, with the exceptions of ConvNeXt and ViT on DomainNet. The same trend holds for in-distribution validation, as seen in fig. C.2.

In summary, while adaptive optimizers can benefit from pre-training in some model architectures, SGD with momentum is the most effective optimization strategy for fine-tuning diverse model architectures. This aligns with and expands upon prior finding (Naganuma et al., 2023) that momentum-based SGD excels for OOD generalization. Our analysis provides practical guidance for fine-tuning optimizer selection.

**Regularization:** We systematically examined the role of regularization, specifically weight decay, when fine-tuning diverse pre-trained models. Our experiments evaluated OOD test accuracy across a range of weight decay coefficients from $1.0 \times 10^{-4}$ to $1.0 \times 10^{-2}$. The results in fig. B.2 reveal differing trends between PACS, OfficeHome, and DomainNet versus VLCS.

For the former group, stronger regularization hurts OOD test accuracy while improving calibration (ECE), as shown in fig. C.1. This represents a trade-off between OOD performance and uncertainty estimates. In contrast, for VLCS, increased weight decay improves both OOD accuracy and ECE.

The discrepancy likely stems from dataset differences—the former group combines visually distinct domains (art, photo, sketch), while VLCS merges similar photo datasets.

To summarize, regularization does not lead to unified conclusions across distribution shifts. While helpful for calibration, weight decay can degrade OOD accuracy on domain shifts with more separated distributions like PACS.

## B.3   Relationship with Other OOD Datasets

While our main experiments use domain generalization datasets like PACS, VLCS, OfficeHome, and DomainNet, OOD generalization also encompasses robustness, adversarial examples, sampling bias, and other distribution shifts. Therefore, we evaluated all our pre-trained models on ImageNet OOD test sets (ImageNet-A, ImageNet-R, ImageNet-v2, and ImageNet-Real) for measuring robustness and sampling bias effect and analyzed how these results relate to domain generalization performance.

Notably, we find a rank correlation between pre-training and fine-tuning performance. Models better on ImageNet OOD test sets also achieve higher domain generalization accuracy after fine-tuning (fig. D.9). Further, the correlation exhibits distinct trends — improving the last few percent on robustness datasets, i.e., ImageNet-R and ImageNet-A, give diminishing returns for domain generalization. However, gains on sampling bias datasets, i.e., ImageNet-v2 and ImageNet-Real, contribute substantially.

These trends reveal nuanced relationships between how models generalize on different OOD datasets. Performance on robustness benchmarks provides valuable signals, but still differs from the tasks casted to domain generalization.

# C   Ablation Study

## C.1   The Effects of Random Seeds

Table C.1 demonstrates the mean and standard deviation of in-domain validation accuracy and out-of-distribution test accuracy of models that achieved the highest validation performance over three runs with different random seeds. Again, only selected models are presented to avoid verbose tables. Notice that the standard deviation of both in-domain validation accuracy and out-of-distribution test accuracy is so small that our conclusion is valid.

Table C.1: Mean and standard deviation of in-domain validation accuracy and out-of-distribution test accuracy. Models are trained with SGD with a learning rate of $1.0 \times 10^{-4}$ and a momentum rate of 0.9 over three different random seeds.

| PACS | | |
|---|---|---|
| Model | In-domain validation accuracy | Out-of-distribution test accuracy |
| ResNet-50 (He et al., 2016) | $96.46 \pm 0.19$ | $82.83 \pm 1.32$ |
| ResNet-101 (He et al., 2016) | $96.85 \pm 0.39$ | $88.17 \pm 0.94$ |
| ResNet-152 (He et al., 2016) | $96.49 \pm 0.64$ | $89.20 \pm 0.52$ |
| ConvNeXt-small (Liu et al., 2022) | $97.69 \pm 0.09$ | $92.60 \pm 0.15$ |
| ConvNeXt-base (Liu et al., 2022) | $97.76 \pm 0.40$ | $93.51 \pm 0.38$ |
| ViT-small (w/ AugReg) (Dosovitskiy et al., 2020; Steiner et al., 2022) | $94.72 \pm 0.18$ | $84.94 \pm 0.08$ |
| ViT-base (w/ AugReg) (Dosovitskiy et al., 2020; Steiner et al., 2022) | $95.70 \pm 0.15$ | $86.99 \pm 0.49$ |
| **OfficeHome** | | |
| Model | In-domain validation accuracy | Out-of-distribution test accuracy |
| ResNet-50 (He et al., 2016) | $88.98 \pm 0.38$ | $60.49 \pm 0.73$ |
| ResNet-101 (He et al., 2016) | $89.48 \pm 0.22$ | $64.86 \pm 0.27$ |
| ResNet-152 (He et al., 2016) | $90.01 \pm 0.45$ | $65.42 \pm 1.18$ |
| ConvNeXt-small (Liu et al., 2022) | $92.28 \pm 0.27$ | $76.15 \pm 0.86$ |
| ConvNeXt-base (Liu et al., 2022) | $92.43 \pm 0.12$ | $77.20 \pm 0.77$ |
| ViT-small (w/ AugReg) (Dosovitskiy et al., 2020; Steiner et al., 2022) | $90.22 \pm 0.39$ | $63.89 \pm 0.97$ |
| ViT-base (w/ AugReg) (Dosovitskiy et al., 2020; Steiner et al., 2022) | $90.90 \pm 0.11$ | $69.86 \pm 0.55$ |
| **DomainNet** | | |
| Model | In-domain validation accuracy | Out-of-distribution test accuracy |
| ResNet-50 (He et al., 2016) | $52.12 \pm 0.36$ | $56.96 \pm 0.31$ |
| ResNet-101 (He et al., 2016) | $53.95 \pm 0.22$ | $58.29 \pm 0.30$ |
| ResNet-152 (He et al., 2016) | $55.44 \pm 0.18$ | $59.84 \pm 0.51$ |
| ConvNeXt-small (Liu et al., 2022) | $63.28 \pm 0.01$ | $68.40 \pm 0.26$ |
| ConvNeXt-base (Liu et al., 2022) | $63.98 \pm 0.13$ | $69.49 \pm 0.43$ |
| ViT-small (w/ AugReg) (Dosovitskiy et al., 2020; Steiner et al., 2022) | $56.03 \pm 0.25$ | $59.07 \pm 0.74$ |
| ViT-base (w/ AugReg) (Dosovitskiy et al., 2020; Steiner et al., 2022) | $59.80 \pm 0.09$ | $64.20 \pm 0.19$ |
| **VLCS** | | |
| Model | In-domain validation accuracy | Out-of-distribution test accuracy |
| ResNet-50 (He et al., 2016) | $85.26 \pm 0.58$ | $73.91 \pm 1.29$ |
| ResNet-101 (He et al., 2016) | $83.69 \pm 0.33$ | $72.72 \pm 2.00$ |
| ResNet-152 (He et al., 2016) | $84.61 \pm 0.62$ | $73.73 \pm 2.05$ |
| ConvNeXt-small (Liu et al., 2022) | $87.45 \pm 0.15$ | $79.61 \pm 0.26$ |
| ConvNeXt-base (Liu et al., 2022) | $87.12 \pm 0.36$ | $77.93 \pm 0.28$ |
| ViT-small (w/ AugReg) (Dosovitskiy et al., 2020; Steiner et al., 2022) | $86.61 \pm 0.50$ | $77.73 \pm 0.22$ |
| ViT-base (w/ AugReg) (Dosovitskiy et al., 2020; Steiner et al., 2022) | $85.68 \pm 0.45$ | $77.74 \pm 0.48$ |

## C.2 The effects of Hyperparameters

Figure C.1 shows expected calibration error rates with respect to model sizes with different weight decay rates of momentum SGD. Unlike out-of-distribution test accuracy in fig. B.2, clear trends cannot be found.

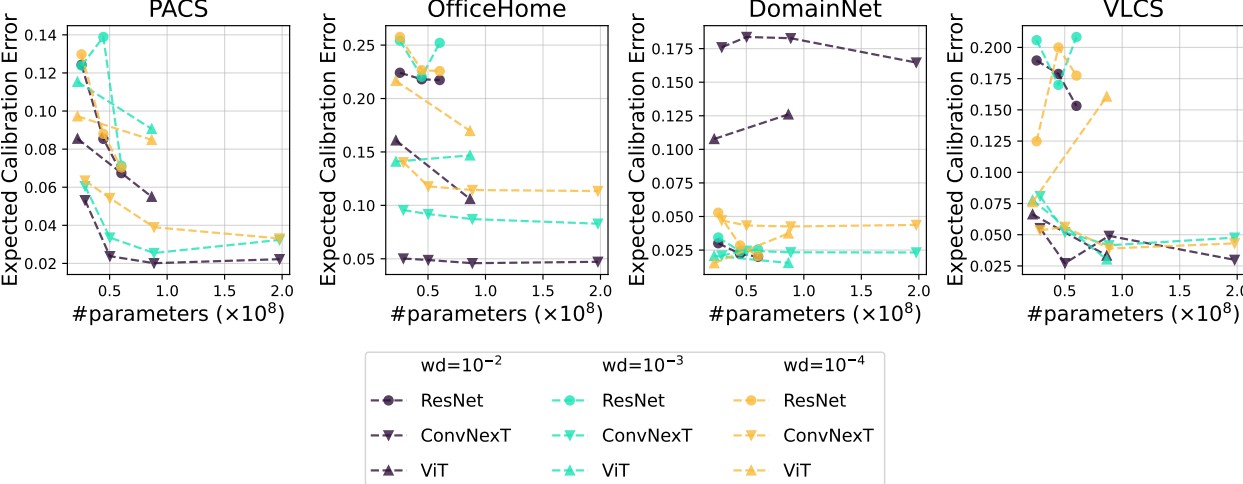

Figure C.1: Out-of-distribution expected calibration error with respect to the number of model parameters with different weight decay rates of momentum SGD.

Figures C.2 and C.3 present the effect of hyperparameter selection on in-domain validation accuracy, which was used to select hyperparameters, and out-of-distribution test accuracy, which was used to report the final performance. These figures only show results of ResNet-50, ConvNeXt-base, ViT-base (w/ AugReg), EfficientNet-B3 to avoid verbosity, and AdamW for ConvNeXt-base is denoted as Adam in fig. C.2 and fig. C.3 for simplicity. Aligned with fig. 2, we can observe that in-domain validation accuracy is a good estimate of out-of-distribution test accuracy, regardless of the model architectures. Also, we notice that SGD with a weight-decay rate of $10^{-4}$ consistently achieves (nearly) the best validation and test accuracy regardless of learning rates.

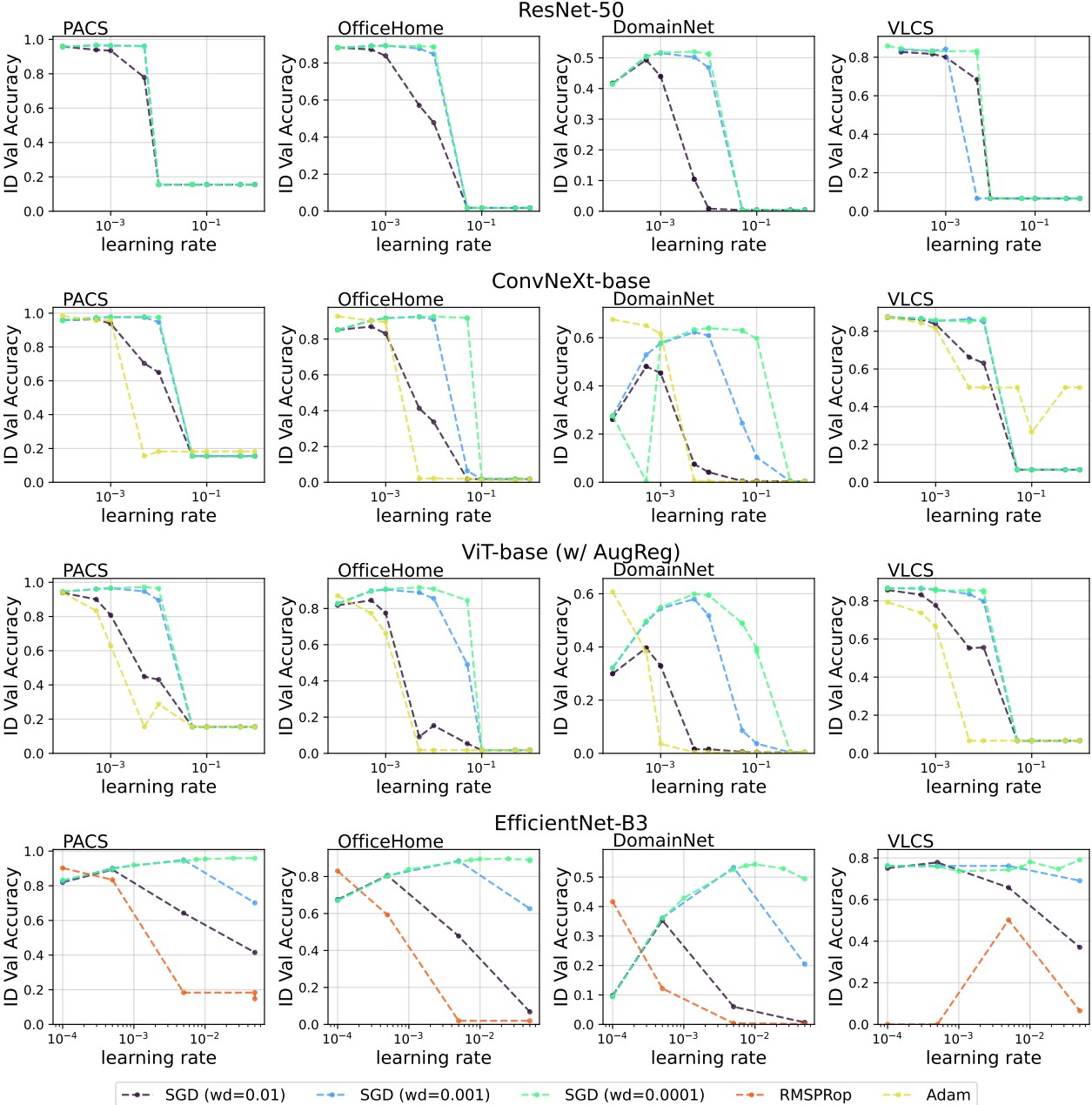

Figure C.2: In-domain validation accuracy with different hyperparameters of optimizers. "wd" stands for the weight decay rate. For simplicity, we convert "NaN" value to 0.

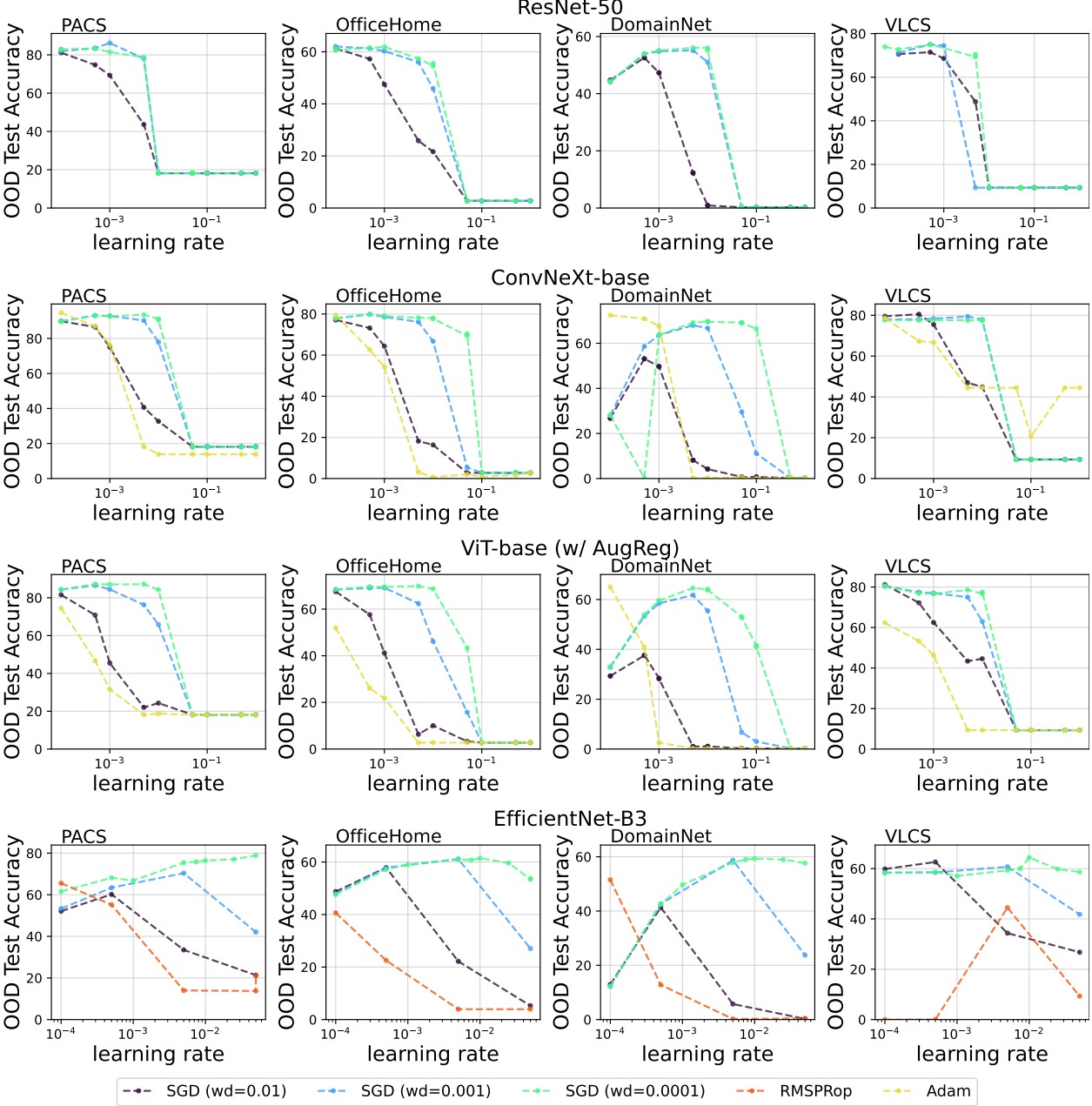

Figure C.3: Out-of-distribution validation accuracy with different hyperparameters of optimizers. "wd" stands for the weight decay rate. For simplicity, we convert "NaN" value to 0.

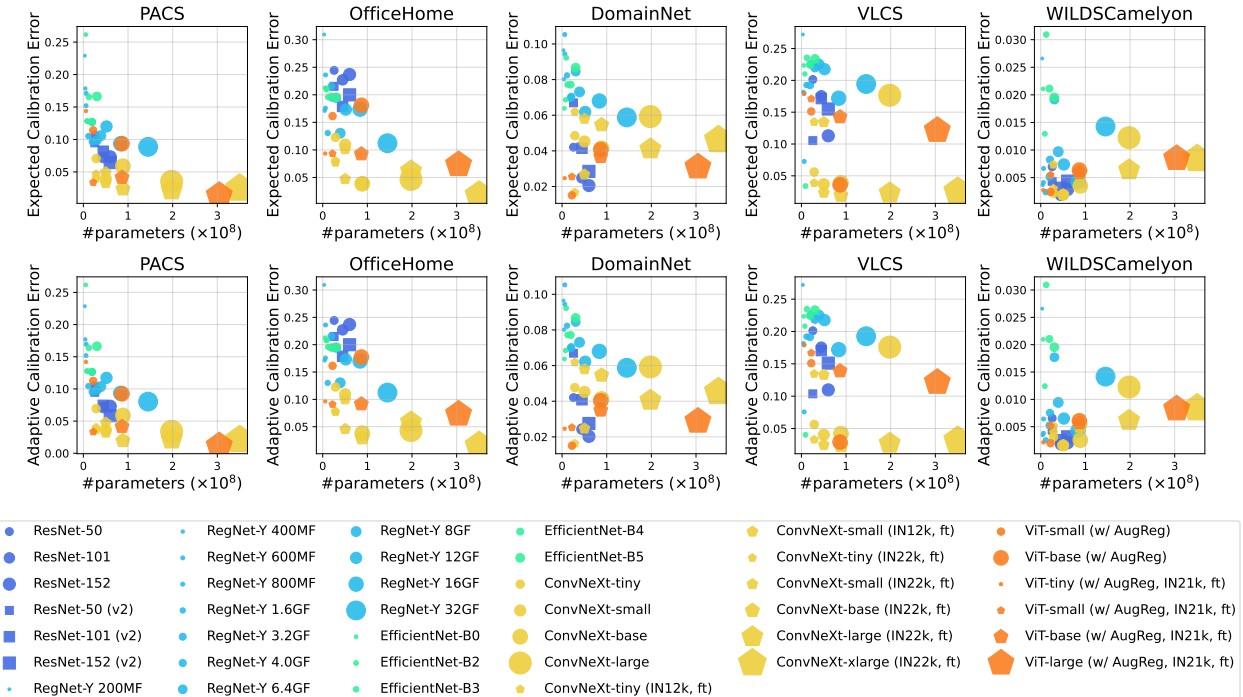

Figure D.1: Expected calibration error (top) and adaptive calibration error (bottom) with respect to the number of parameters.

# D    Additional Results

## D.1    Expected Calibration Error vs Adaptive Calibration Error

Figure D.1 compares expected calibration error (ECE) and adaptive calibration error (ACE). From this figure, we can confirm that ECE and ACE show almost the same values within the scope of our experiments across multiple model architectures and multiple datasets.

## D.2    TerraIncognita Dataset Results

Figure D.2 shows OOD test error, ECE, and ACE on the Terraincognita dataset with the test environment of "L38". Similar trends to the other datasets can be observed.

## D.3    Choice of Test Environments

Figures D.3 and D.4 show OOD test error and ECE evaluated on OfficeHome and PACS with different test environments. We can observe similar trends regardless of the choice of test environments. The reason that the behaviors of OfficeHome (test env = Product) and PACS (test env = Photo) look different from others at a glance, e.g., ConvNeXt-xlarge yields poorer performance than smaller ones on OfficeHome (test env = Product), could be attributed to the saturation of the performance in these environments (see the error rates).

## D.4    Full Results of DomainBed

Figure D.6 presents the relationship between the number of model parameters and out-of-distribution test accuracy (top) and expected calibration error (bottom) with all models considered in this work. This figure is the full version of fig. 1.

Figure D.5 shows the OOD test error and expected calibration error with respect to the number of parameters on all four datasets, corresponding to the left panels of fig. 3.

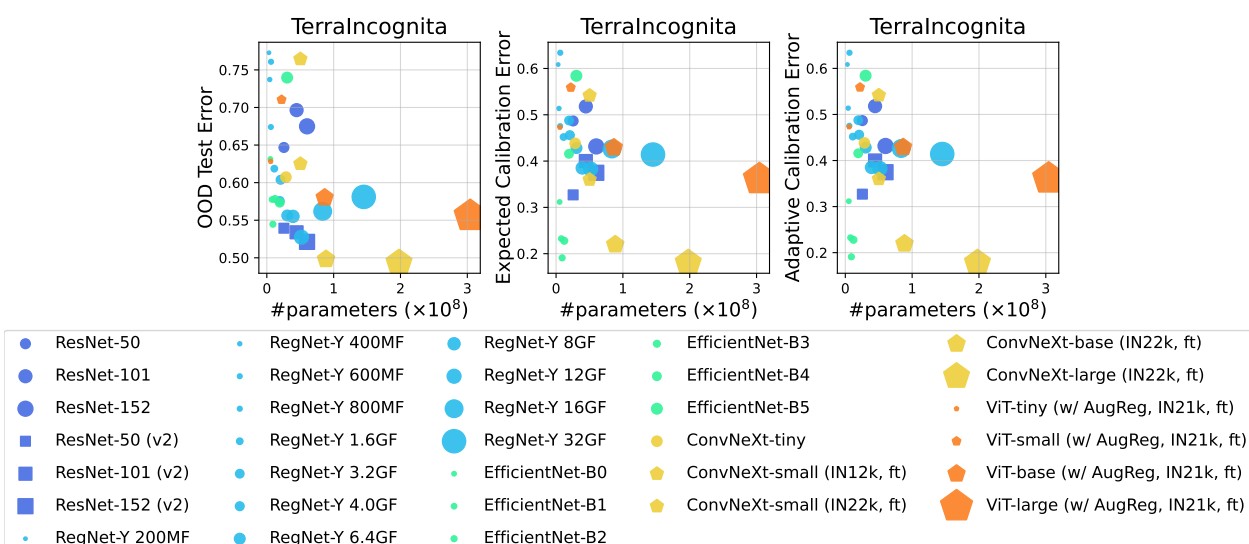

Figure D.2: OOD test error and calibration error metrics on TerraIncognita.

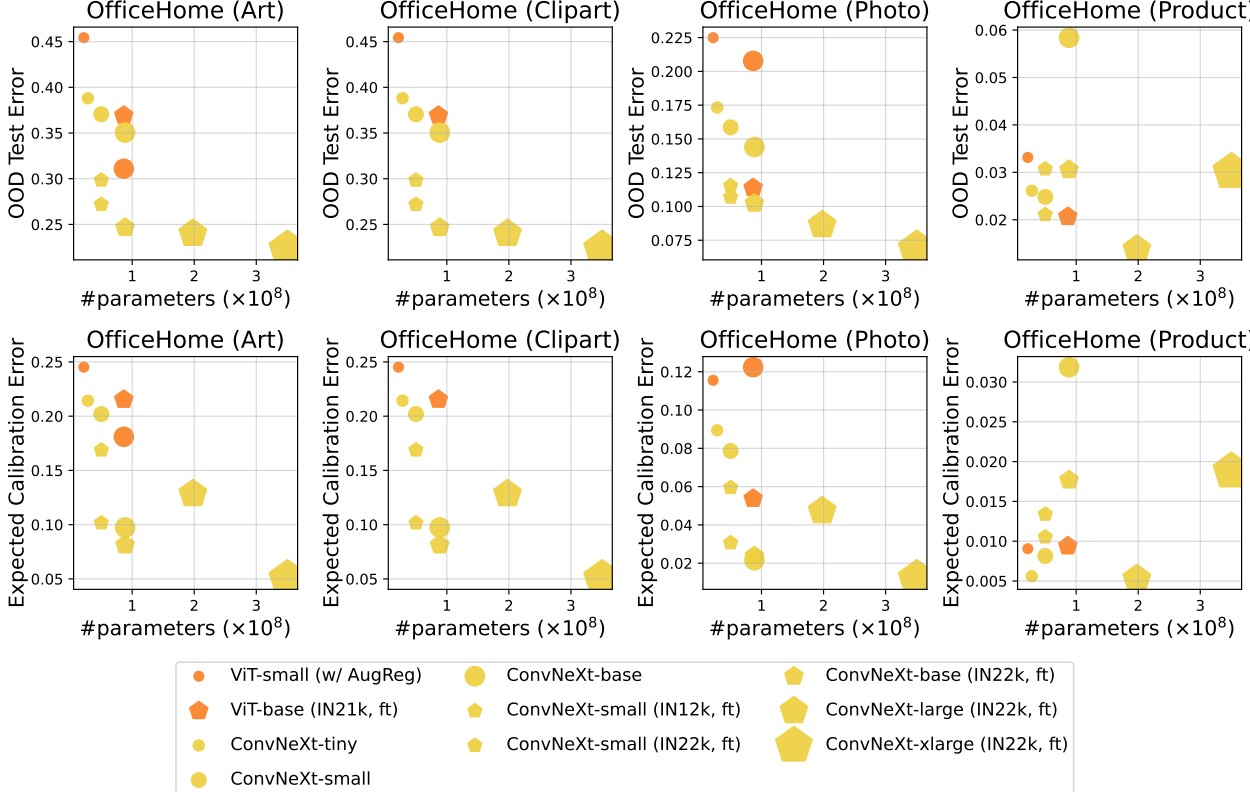

Figure D.3: OOD test error (top) and expected calibration error (bottom) evaluated on OfficeHome with different test environments.

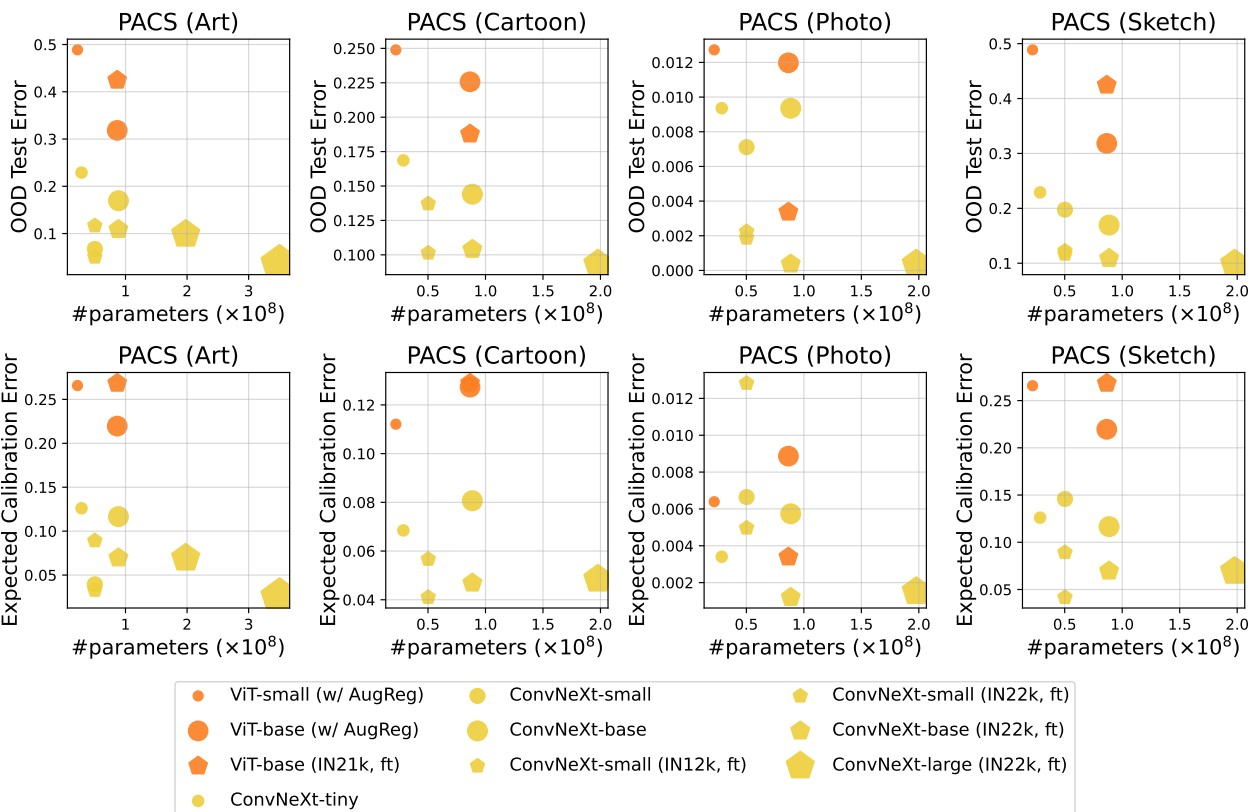

Figure D.4: OOD test error (top) and expected calibration error (bottom) evaluated on PACS with different test environments.

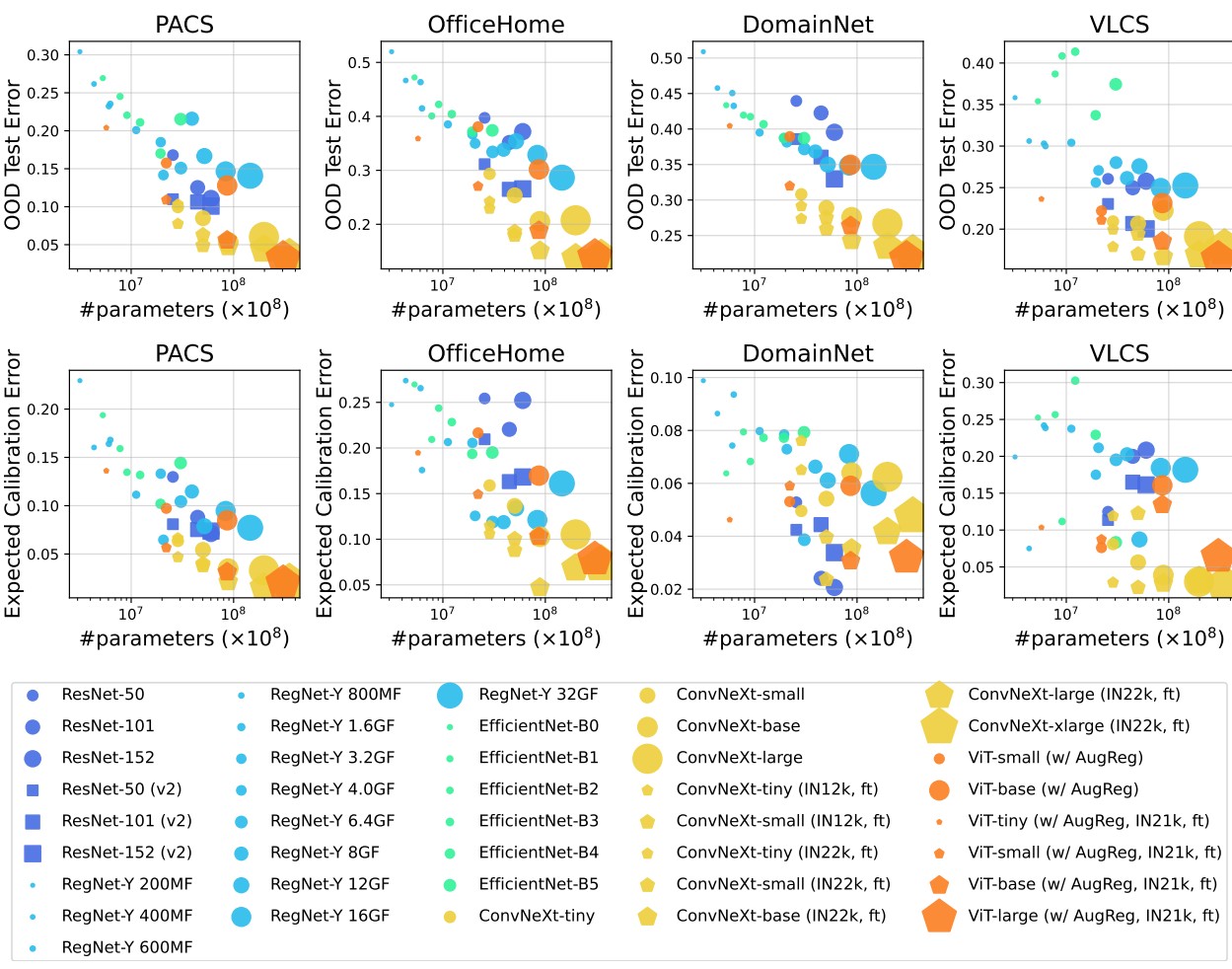

Figure D.5: Out-of-distribution test error (top) and expected calibration error (bottom) with respect to the number of parameters. This figure plots all models used in the experiments.

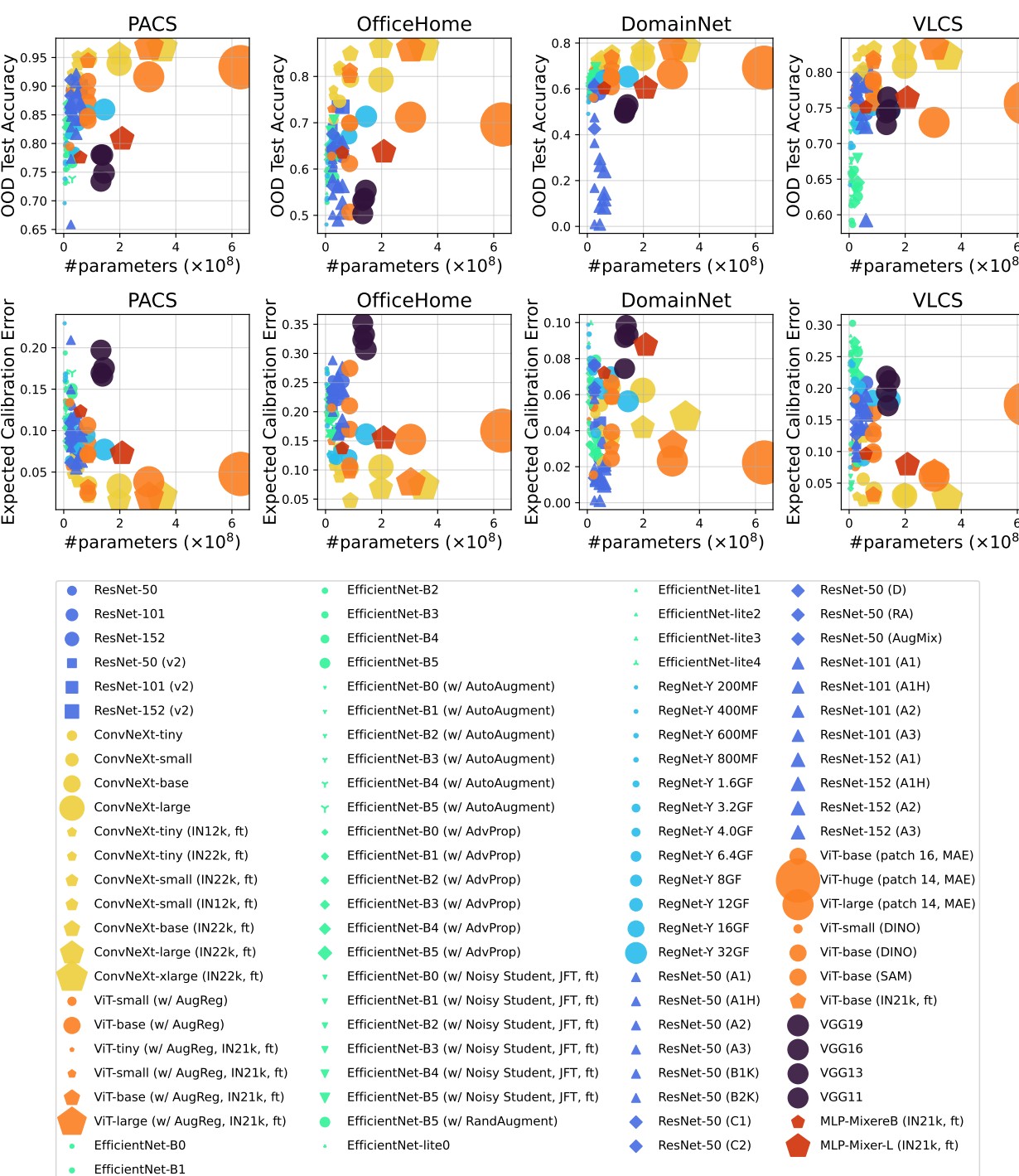

Figure D.6: Out-of-distribution test accuracy (top) and expected calibration error (bottom) with respect to the number of parameters. This figure plots all models used in the experiments.

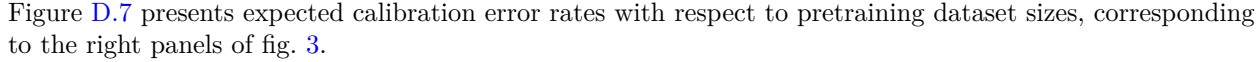

Figure D.7 presents expected calibration error rates with respect to pretraining dataset sizes, corresponding to the right panels of fig. 3.

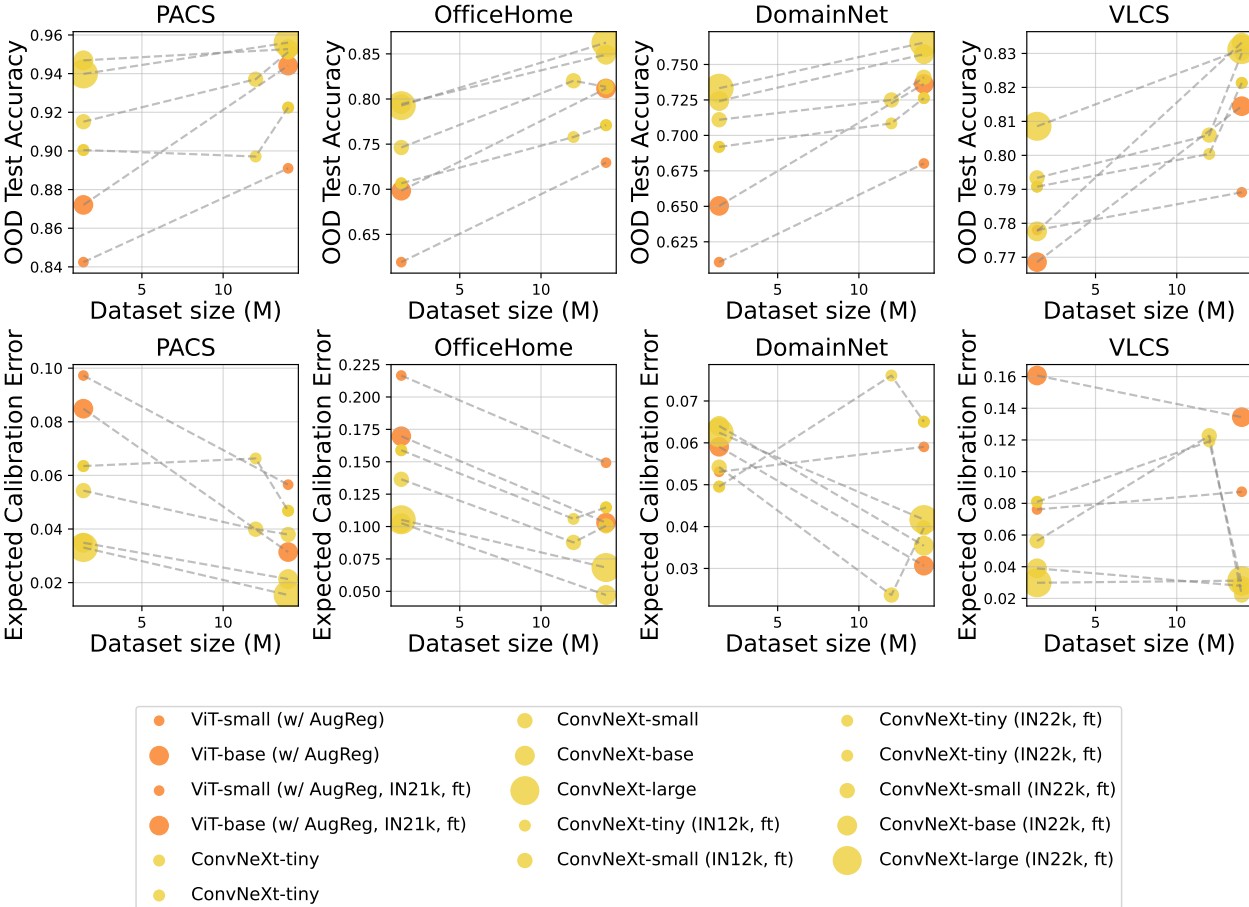

Figure D.7: The relationship between pre-training dataset sizes and expected calibration error (lower is better). The same model architectures with different pre-training datasets are connected by dashed lines.

Figure D.6 is the version of fig. D.5.

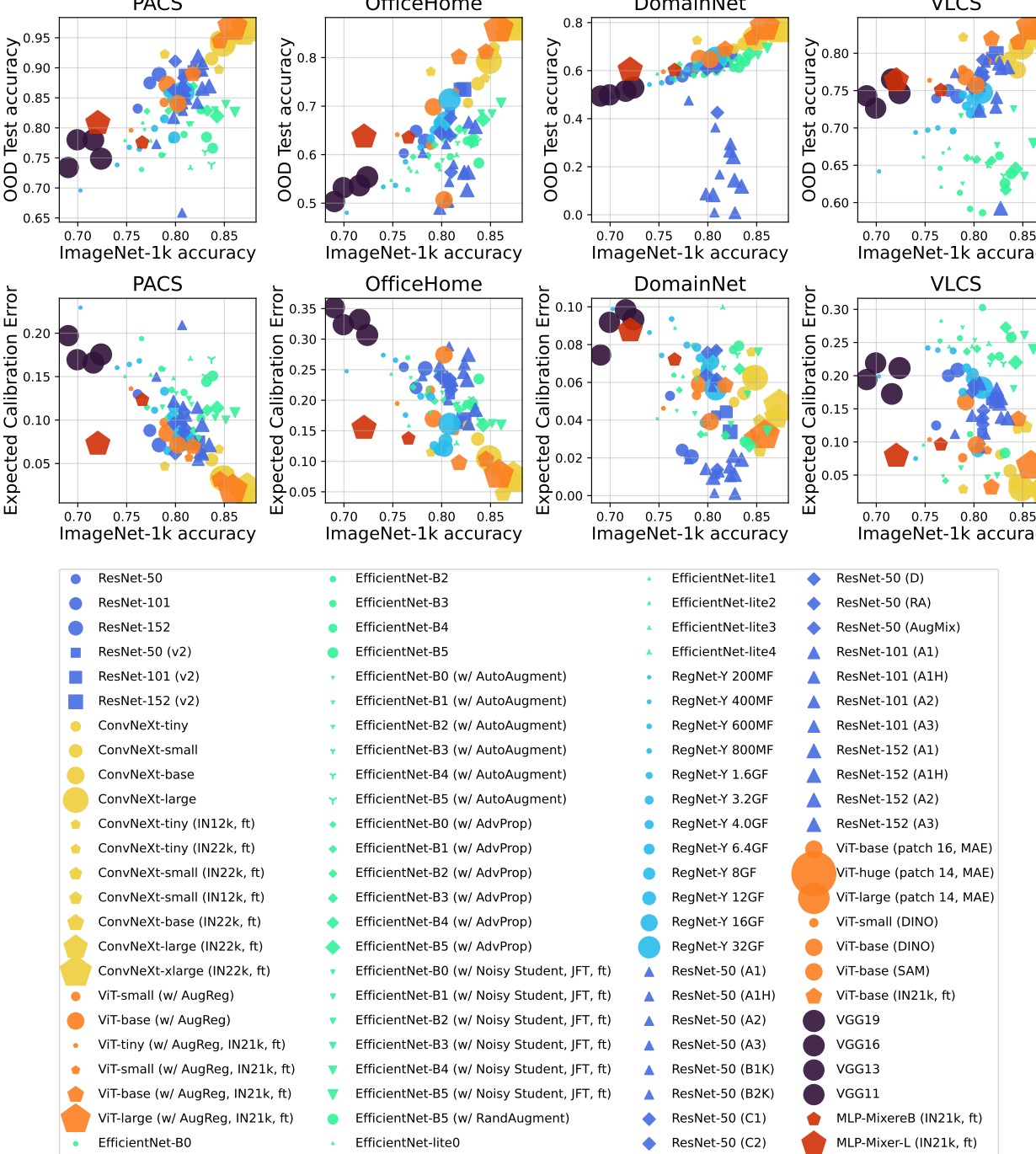

Figure D.8: The relationship of out-of-distribution test accuracy (top) and expected calibration error (bottom) with ImageNet-1k accuracy. This figure plots all models used in the experiments.

Figure D.9 shows the relationship between out-of-distribution test accuracy after finetuning and ImageNet-1k OOD test sets, namely, ImageNet-A (Hendrycks et al., 2021b), ImageNet-R (Hendrycks et al., 2021a), ImageNet-Real (Beyer et al., 2020), and ImageNet v2 (Recht et al., 2019), before finetuning.

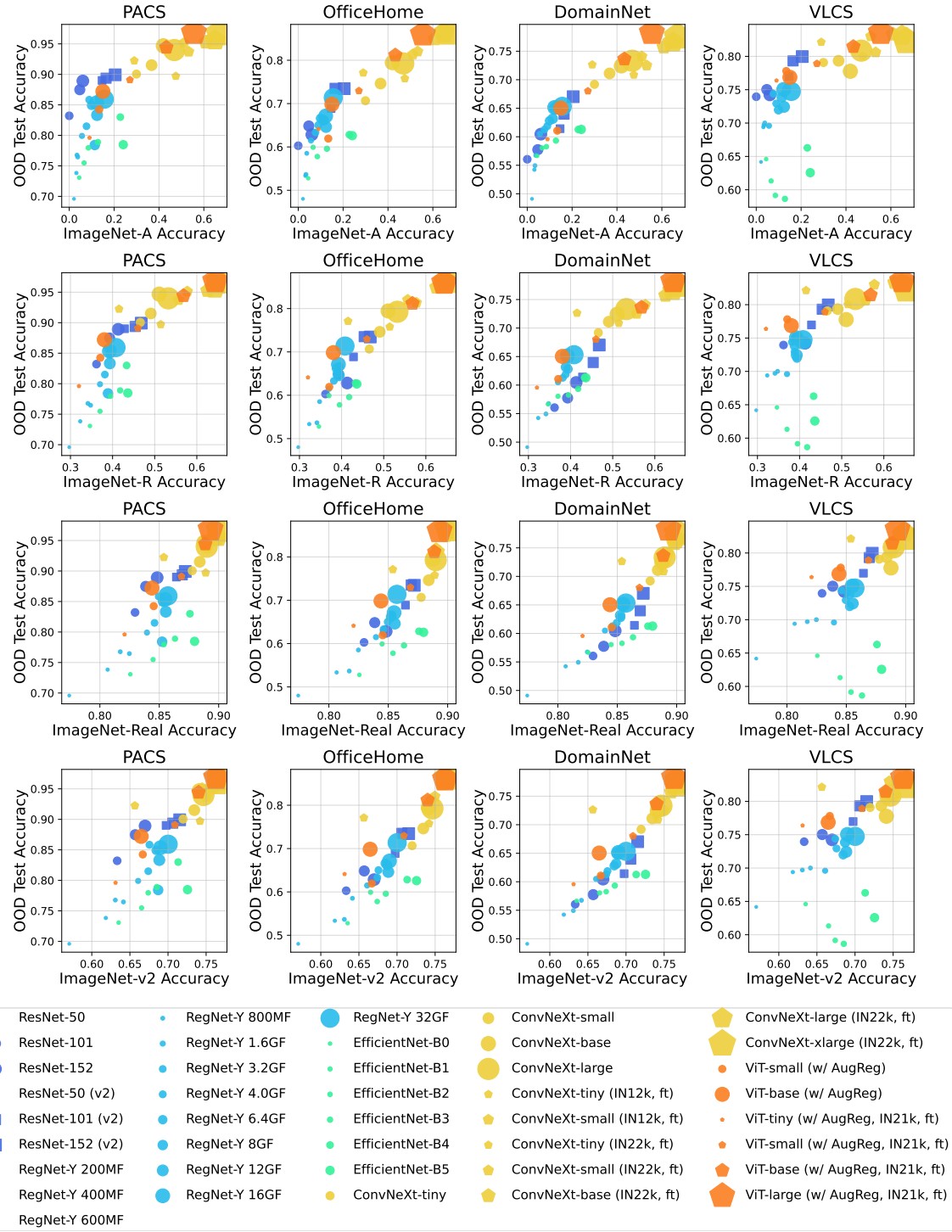

Figure D.9: Relationship between out-of-distribution test accuracy after finetuning and ImageNet-1k OOD test sets before finetuning.

Figure D.10 displays the relationship between accuracy on ImageNet-1k validation set and ImageNet OOD test sets.

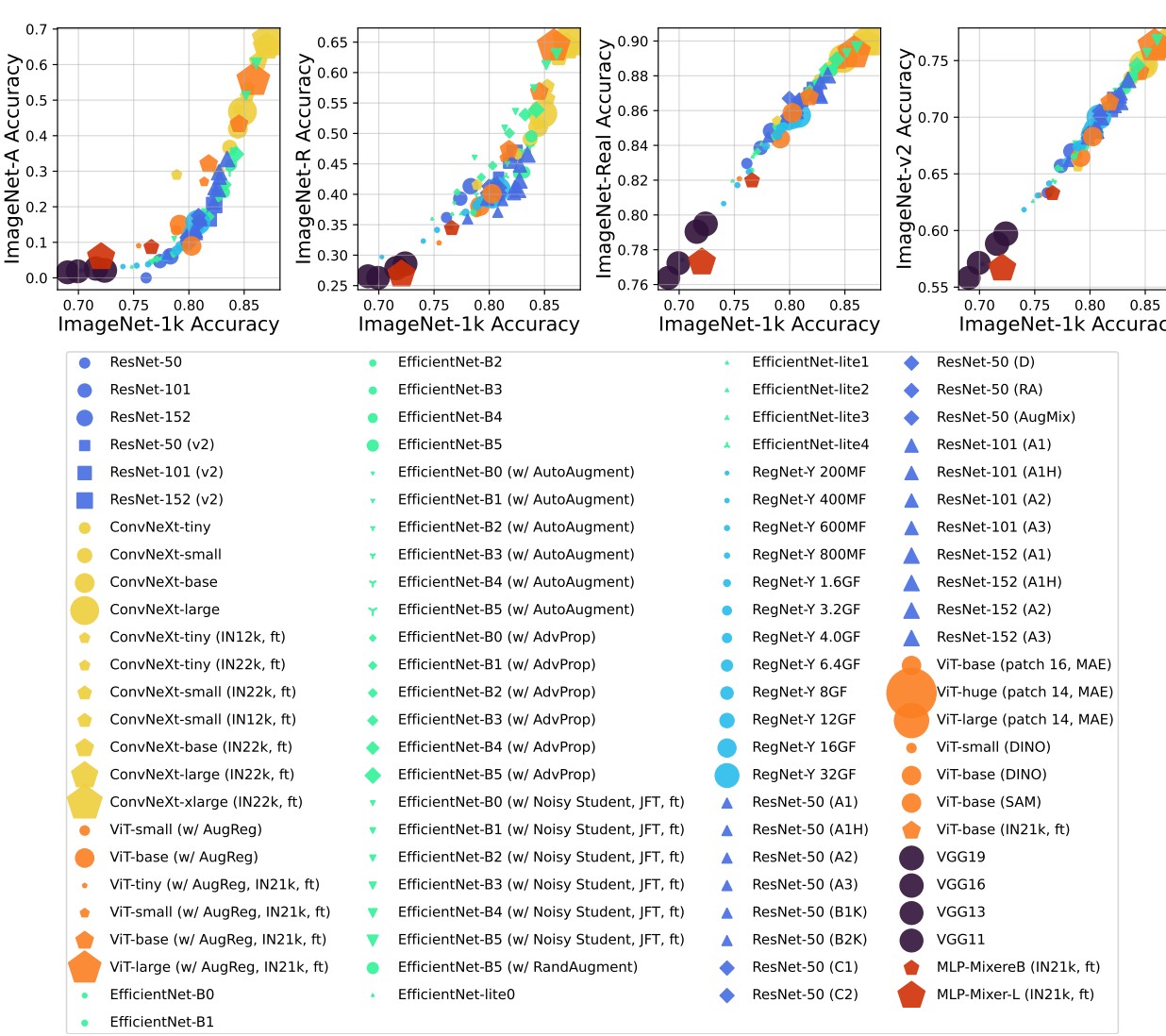

Figure D.10: The relationship between ImageNet-1k validation accuracy and accuracy on ImageNet OOD test sets.

## D.5    Results per Model Architecture

Figures D.11 and D.12 show OOD test accuracy and expected calibration error rates of EfficientNets with various training schemes. Different from out-of-distribution test accuracy, data augmentation (AutoAugment and AdvProp) or additional unlabeled data (NoisyStudent on JFT) do not always contribute to ECE improvement.

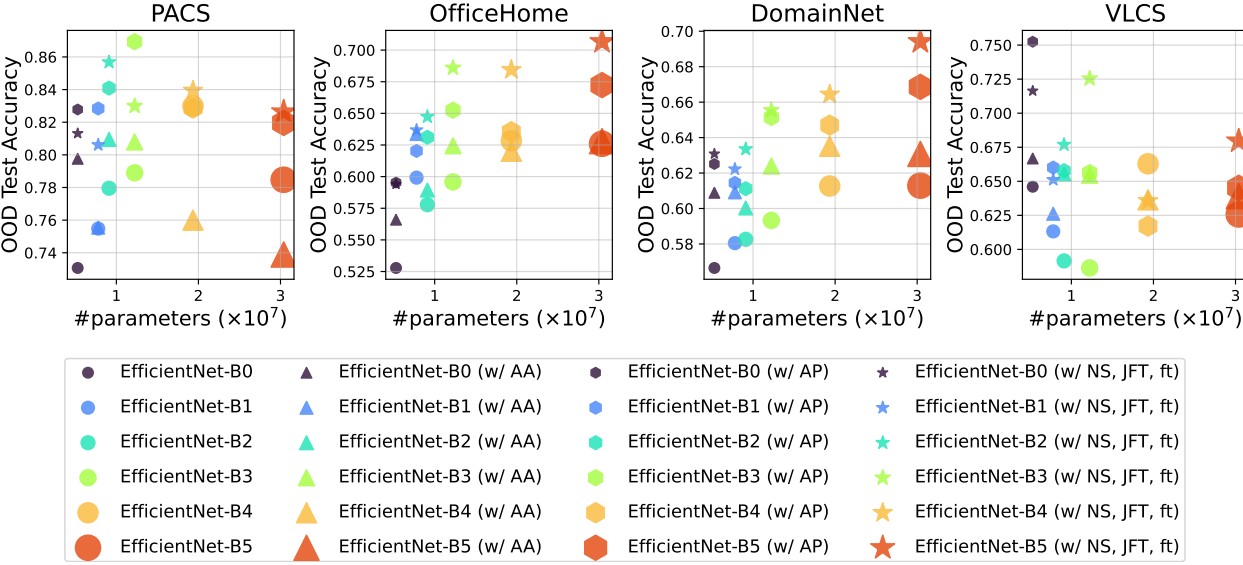

Figure D.11: Out-of-distribution test accuracy of EfficientNets (Tan & Le, 2019) with various training schemes. AA, AP, and NS stand for AutoAugment (Cubuk et al., 2019), AdvProp (Xie et al., 2020a), and Noisy Student (Xie et al., 2020b), respectively. JFT is Google's internal unlabeled dataset consisting of 300 million images.

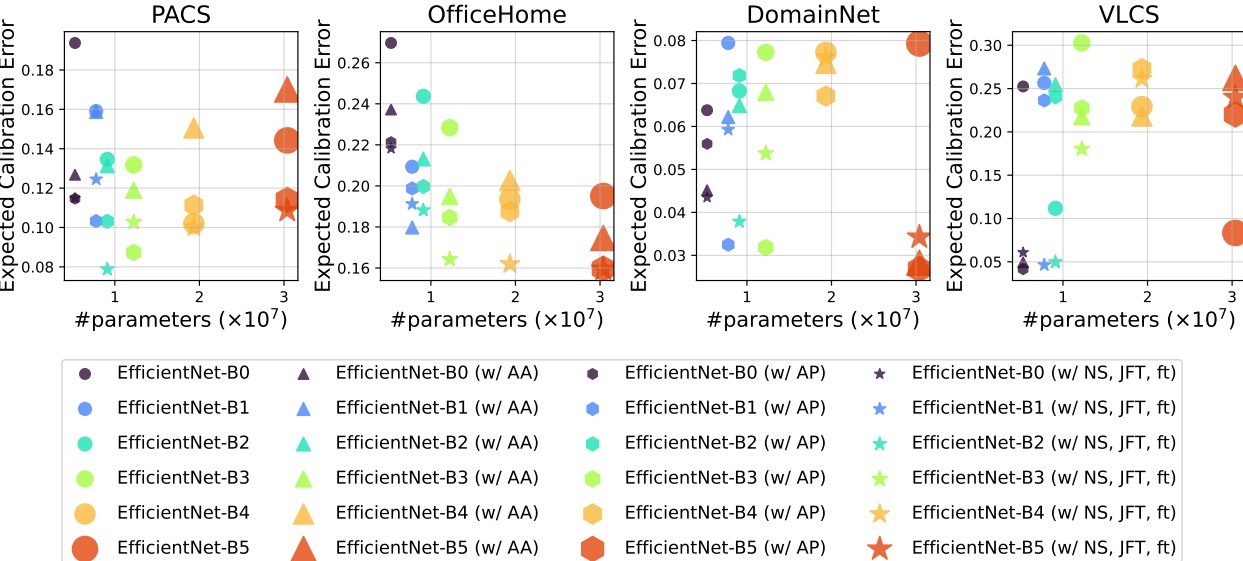

Figure D.12: Out-of-distribution expected calibration error of EfficientNets (Tan & Le, 2019) with various training schemes. AA, AP, and NS stand for AutoAugment (Cubuk et al., 2019), AdvProp (Xie et al., 2020a), and Noisy Student (Xie et al., 2020b), respectively. JFT is Google's internal unlabeled dataset consisting of 300 million images.

Figure D.13 demonstrates expected calibration error rates with respect to model sizes of ResNets with different pre-training schemes (Wightman et al., 2021). Similar to their out-of-distribution test accuracy in fig. 5, extensive pre-training schemes (A1-A3) do not always contribute to the ECE improvement.

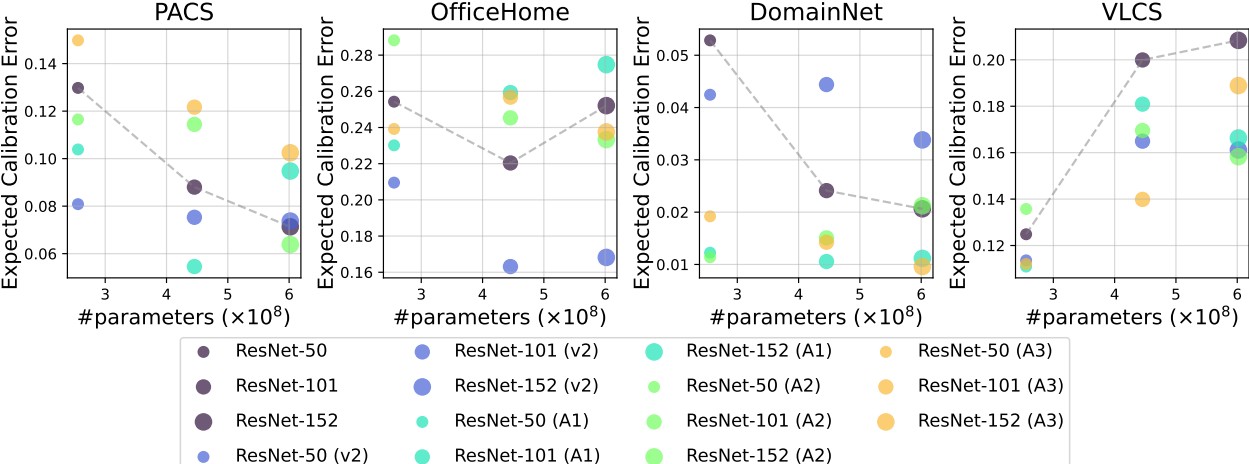

Figure D.13: The relationship between model sizes and expected calibration error (lower is better) of ResNets with different pre-training schemes (Wightman et al., 2021). The original ResNet models are connected by dashed lines.

Figure D.14 shows the relationship between ImageNet-1k validation accuracy and OOD test accuracy of ResNet-50, pre-trained with various recipes, including RandAugment (Cubuk et al., 2020) and AugMix (Hendrycks* et al., 2020).

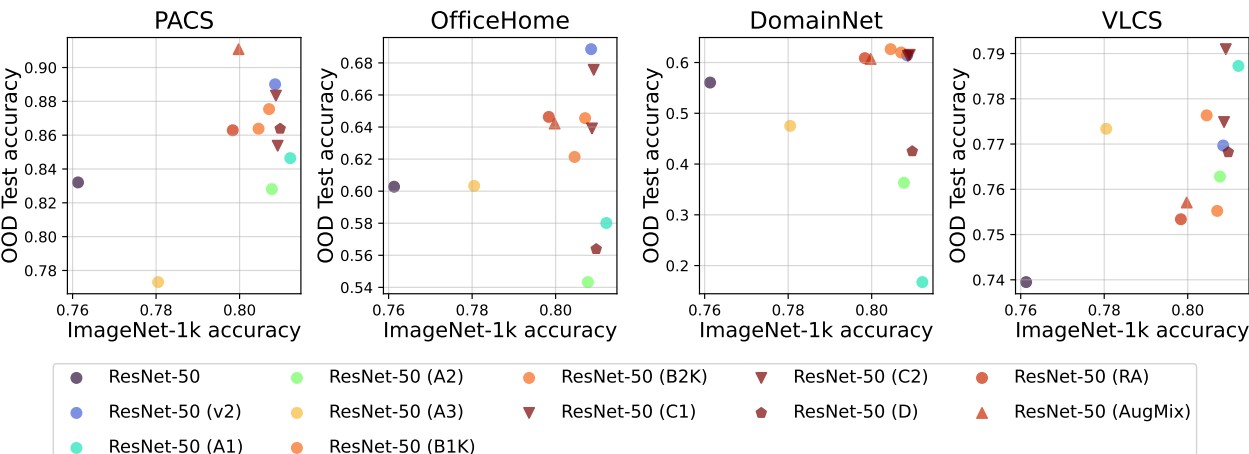

Figure D.14: The relationship between ImageNet-1k validation accuracy and out-of-distribution test accuracy (higher is better) of ResNet-50 with different pre-training schemes, such as (Wightman et al., 2021).

Figure D.15 shows OOD test accuracy with respect to ImageNet-1k validation accuracy of Vision Transformers, pre-trained in supervised learning.

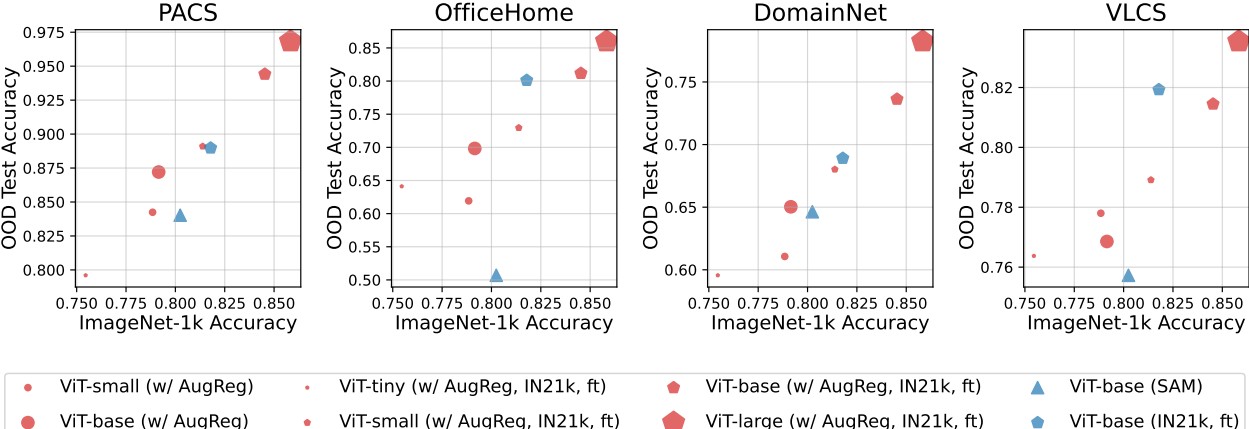

Figure D.15: The relationship between ImageNet-1k validation accuracy and out-of-distribution of Vision Transformers trained in supervised learning. SAM is an optimizer, standing out for Sharpness Aware Minimization (Foret et al., 2021).

Figure D.16 shows the expected calibration error of Vision Transformers with various training schemes, including self-supervised learning, namely DINO (Caron et al., 2021) and MAE (He et al., 2022), corresponding to fig. B.1.

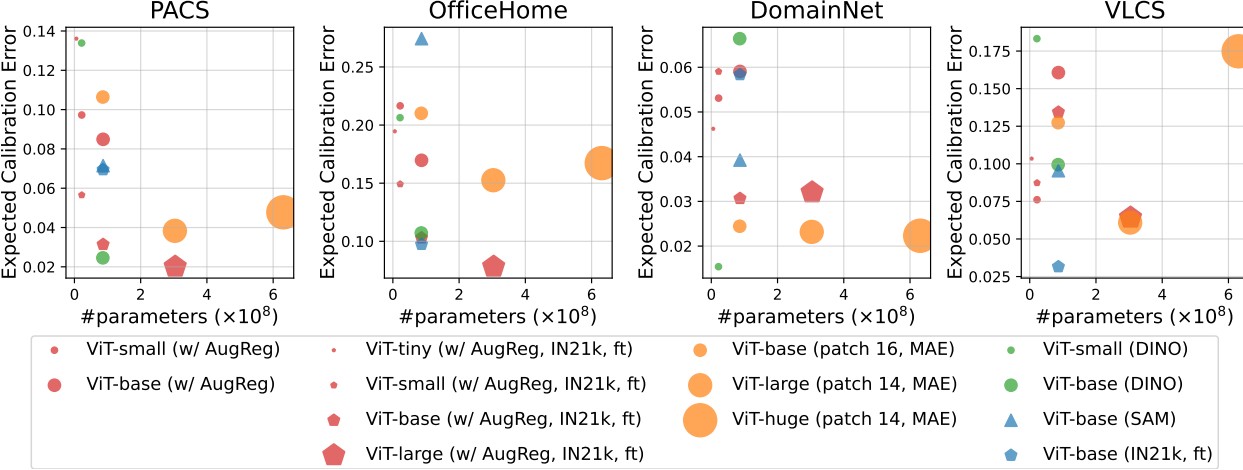

Figure D.16: Out-of-distribution expected calibration error of Vision Transformers (Dosovitskiy et al., 2020) with various training schemes. Comparing the same architectures aligned in vertical lines, supervised pre-training outperforms self-supervised pre-training (DINO (Caron et al., 2021) and MAE (He et al., 2022)). Also interestingly, pre-training with the SAM optimizer (Foret et al., 2021) yields inferior performance to its SGD counterpart.

## D.6 Results of Camelyon17 Datasets

Figure D.17 presents the OOD test accuracy and the expected calibration error on Camelyon17, consisting of medical images that are completely different from the domains of the pre-training data. As the task is relatively easier than others, all models can achieve over 90% test accuracy and near 0 ECE results, which makes it difficult to differentiate the characteristics of the models. Yet, we can still observe the general trend that larger models yield better OOD test accuracy and ECE, supporting our claims.

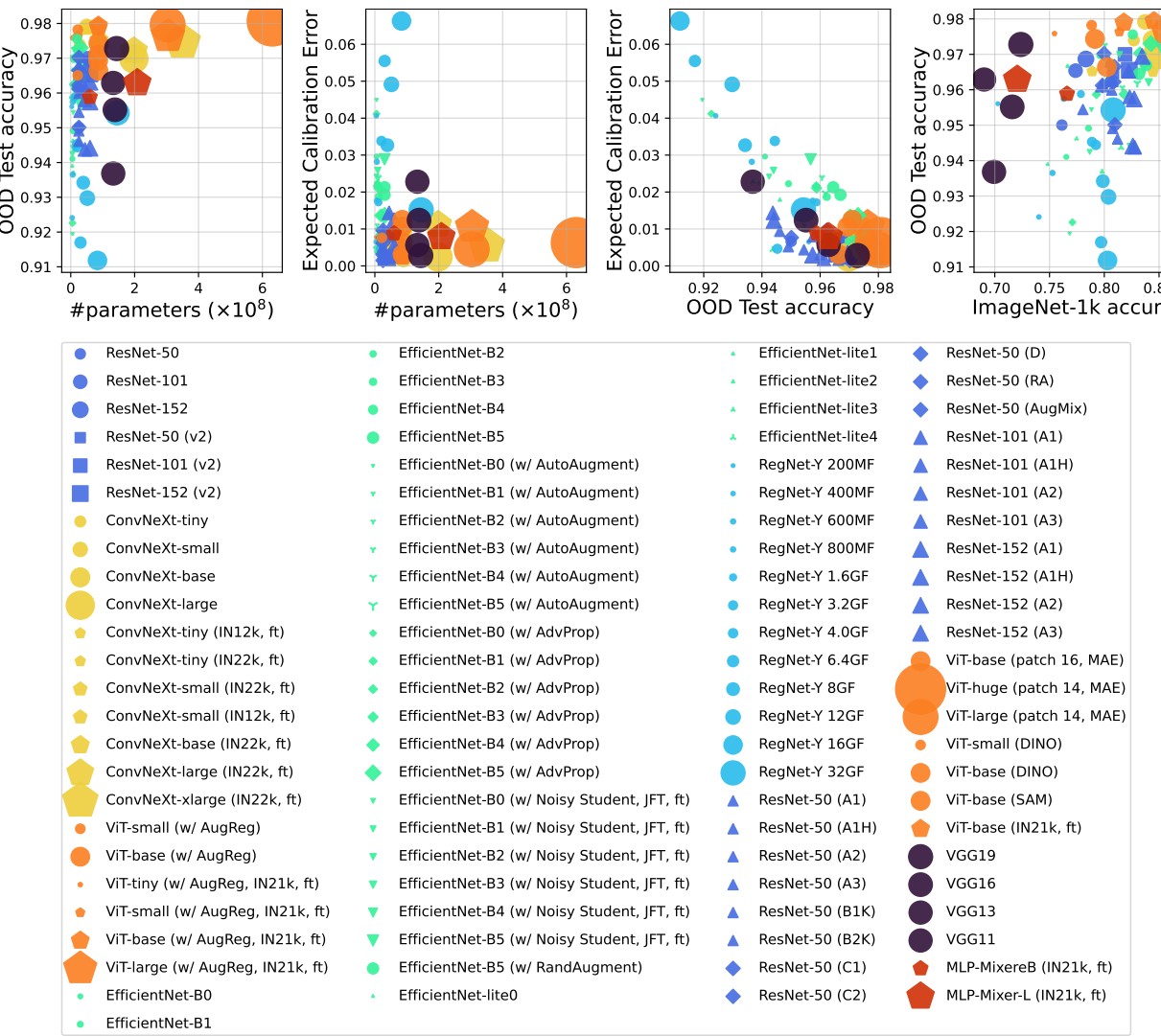

Figure D.17: OOD test accuracy, expected calibration error, and their relationship with the number of model parameters and ImageNet-1k accuracy..

### D.7 Results of CLIP models

As a reference, fig. D.18 shows the test accuracy and expected calibration error in the zero-shot CLIP models with different visual encoders. We can observe that CLIP models achieve nearly comparable accuracy to other models, while they degenerate calibration scores as the number of parameters in the visual encoder increases.

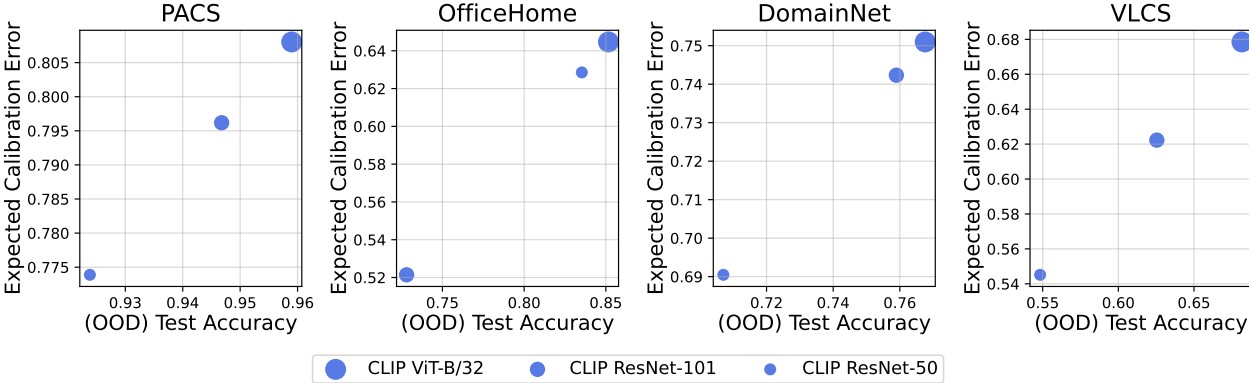

Figure D.18: The relationship between (out-of-distribution) test accuracy and expected calibration error of CLIP models with different visual encoders.

## E    ECE Under Infinite Data and Near Zero Cross-Entropy Loss

In this section, we provide **Theorem E.1.**, which shows that, given an infinite dataset and a sufficiently parameterized model, the Expected Calibration Error (ECE) converges to zero.

**Theorem E.1** (informal)**.** *Let $p(y|x)$ denote the model's predicted probability distribution over the target variable $y$ given input $x$, and let $P(y|x)$ represent the true conditional probability distribution. Suppose the following conditions hold:*

*1. The dataset is infinite, and the empirical data distribution matches the true distribution*

*2. The model achieves near-zero cross-entropy loss on this dataset.*

*Then, under these assumptions, the model's predictions $p(y|x)$ converge to the true distribution $P(y|x)$, resulting in an ECE that approaches zero.*

*Proof.* To start, we define the cross-entropy loss for a classifier with predicted probabilities $p(y|x)$ and true labels $y$ drawn from the data distribution:

$$L = -\mathbb{E}_{(x,y)\sim P_{\text{data}}}\left[\log p(y|x)\right].$$

With an infinite dataset, the empirical data distribution aligns with the true distribution, so the empirical average of the loss matches the expected value.

Given that the cross-entropy loss $L$ is nearly zero, we can infer that $p(y|x)$ is close to $P(y|x)$, as cross-entropy loss minimizes when $p(y|x) = P(y|x)$. To clarify, we can express the cross-entropy loss as:

$$L = H(P(y|x)) + \text{KL}(P(y|x)\|p(y|x)),$$

where $H(P(y|x))$ is the entropy of the true distribution, and $\text{KL}(P(y|x)\|p(y|x))$ represents the Kullback-Leibler (KL) divergence between $P(y|x)$ and $p(y|x)$. Since the KL divergence is non-negative and zero only when $p(y|x) = P(y|x)$, a near-zero cross-entropy loss suggests that $p(y|x) \approx P(y|x)$.

Now, considering the ECE, if $p(y|x) \approx P(y|x)$, each predicted probability closely approximates the true conditional probability. This leads to the following relation:

$$
\begin{aligned}
\mathrm{acc}\,(B_m) &= \frac{1}{|B_m|} \sum_{i \in B_m} \mathbf{1}\,(\hat{y}_i = y_i) \\
&\approx \mathbb{E}_{(x,y) \sim P_{\mathrm{data}}} \left[ P(y|x) \,|\, p(y|x) \in B_m \right] \\
&\approx \mathbb{E}_{(x,y) \sim P_{\mathrm{data}}} \left[ p(y|x) \,|\, p(y|x) \in B_m \right] \\
&\approx \mathrm{conf}\,(B_m).
\end{aligned}
$$

Refer to Section 2 for notation definitions. Thus, for each bin $B_m$, the predicted confidence $\mathrm{conf}(B_m)$ and true accuracy $\mathrm{acc}(B_m)$ are approximately equal.

Using this in the ECE formula, we have:

$$
\mathrm{ECE} = \sum_{m=1}^{M} \frac{|B_m|}{N} \left| \mathrm{acc}(B_m) - \mathrm{conf}(B_m) \right| \approx 0.
$$

Therefore, under the conditions of infinite data and near-zero cross-entropy loss, we conclude that ECE $\to$ 0. $\qquad\square$

