# OpenReview forum: "An Empirical Study of Pre-trained Model Selection for Out-of-Distribution Generalization and Calibration"
_TMLR — Accepted by TMLR_

### Review · Reviewer_kFXi · 2024-11-17

**Summary Of Contributions:**

This paper studies the relation between model size and accuracy and calibration quality in computer vision. It runs extensive experiments across models and datasets, and summarize the ECE and accuracy as models grow. It also studies the effect of pre-training on these evaluation metrics.

**Audience:**

Yes

**Claims And Evidence:**

No

**Requested Changes:**

1. Please justify the language about uncertainty quantification, or change it to be the more appropriate ``confidence calibration'' throughout.

2. Please address the concerns about ECE (e.g. report other measures such as adaptive ECE, kernel based ECE, class ECE, etc.)

3. Please limit the scope to classification in computer vision tasks.

**Strengths And Weaknesses:**

## Strengths

The strength of this paper is clearly its extensive experiment. There are some interesting observations in Figure 3. The conclusions are intuitive.

## Weaknesses

1. I disagree that ECE should be considered an uncertainty quantification metric. If the authors believe that it is "widely used'' please include references. It is widely used in calibration literature, and calibration has a connection to uncertainty quantification, but they are still very different. For one, calibration is inherently a very frequentist concept yet most UQ works (various BNNs, MCMC, GP) have a Bayesian perspective. Please use the term "confidence calibration" instead.

2. In fact I'd argue ECE's not even a good measure for calibration anymore. This paper didn't specify how the bins for ECE are constructed, so I assume it follows the practice of (Guo et al. 2017) which is using bins of equal width. This is particularly problematic given the nature of this paper, which compares models with very different size and accuracy. This is because ECE takes the average of predictions within a bin.

    a. To see this, for simplicity assume we have a binary classification problem with 50% cat and 50% dog. A dummy classification that always predict $\hat{p}=[0.5,0.5]$ is perfectly calibrated. Now, consider a high-accuracy classifier that tend to predict only in the bin [0.95, 1]. A 20-bin ECE, for the same reason as before, is essentially degenerate for this classifier.

    b. There are various improved measurements of calibration. At the very least, please use adaptive bins (e.g. [0%,5%] percentile of the prediction).

    c. On a side note, ECE only measures *confidence calibration* which is also quite outdated, as it ignores the non-maximum predictions. I suggest going through [1-3] as a start to see why ECE is problematic.

3. Also, the paper uses only computer vision datasets and models, relying heavily on the timm library. Thus, I think the current title and abstract is too general.



[1] Nixon, Jeremy, Michael W. Dusenberry, Linchuan Zhang, Ghassen Jerfel, and Dustin Tran. "Measuring Calibration in Deep Learning." In CVPR workshops, vol. 2, no. 7. 2019.

[2] Widmann, David, Fredrik Lindsten, and Dave Zachariah. "Calibration tests in multi-class classification: A unifying framework." Advances in neural information processing systems 32 (2019).

[3] Kull, Meelis, Miquel Perello Nieto, Markus Kängsepp, Telmo Silva Filho, Hao Song, and Peter Flach. "Beyond temperature scaling: Obtaining well-calibrated multi-class probabilities with dirichlet calibration." Advances in neural information processing systems 32 (2019).

---

> ### Author Response · Authors · 2025-01-31
> **Response to reviewer kFXi**
>
> We sincerely appreciate for your review of our manuscript. We greatly appreciate your insights, and we have carefully incorporated your suggestions into our revised manuscript. Below, we present our point-by-point responses to your comments.
>
> __(W1) Term of uncertainty quantification:__
>
> We agree that linking uncertainty quantification with ECE could cause confusion.
> Therefore, we have revised the terminology in Section 2: Preliminary, replacing uncertainty quantification with confidence calibration.
>
> __(W2) ACE vs ECE:__
>
> As you pointed out, our evaluation of ECE used equal-width binning, which we agree [1] claimed some limitations in certain cases.
> To address this, we comprehensively compared both metrics of ECE and ACE on our experimental protocol (see figure D.1 and D.2).
> Our results showed no significant difference in trends between ECE and ACE, leading to the same conclusions. This further reinforces the observations we initially reported.
>
> At the time of our paper submission, the optimal hyperparameters had already been identified by extensive hyperparameter search, allowing us to somewhat reduce the computational cost of recalculating ACE. However, this process still required significant resources—approximately 4,000 GPU hours in total (equivalent to $12,240 on AWS V100 (P3.8xlarge)), given 100 models × 5 datasets × approx 8 hours. With our 4-GPU cluster, this computation took approximately one and a half months.
>
> __(W3) Scope of our work:__
>
> After revision, we have explicitly clarified in the Abstract, Introduction, and Conclusion that our study is limited to the scope of computer vision.
>
>
> [1] Jeremy Nixon, Michael W Dusenberry, Linchuan Zhang, Ghassen Jerfel, and Dustin Tran. Measuring calibration in deep learning. In CVPR workshops, volume 2, 2019.

---

### Review · Reviewer_AqLf · 2025-01-13

**Summary Of Contributions:**

This article starts with the fine-tuning of pre-trained models, exploring the impact of factors such as the size of pre-trained models, the size of pre-trained datasets, and training strategies on downstream task generalization and uncertainty calibration. Through extensive experiments, it makes some findings, including the confirmation that the choice of pre-trained models indeed affects downstream tasks.

**Audience:**

Yes

**Broader Impact Concerns:**

No.

**Claims And Evidence:**

Yes

**Requested Changes:**

See above.

**Strengths And Weaknesses:**

Strengths
1. The authors conducted a lot of experiments.

Weaknesses
1. The contribution to academia is relatively limited. The work resembles more of a pre-trained model training manual rather than presenting generalizable or quantifiable conclusions.
2. Could the study incorporate some newer architectures, such as Mamba or recent Vision-Language Models (VLMs)?
3. The selection of datasets lacks diversity. OfficeHome, VLCS, and PACS belong to the same dataset series. It would be more compelling to include experiments on more diverse datasets, such as Woods.
4. Using the first domain of each dataset as the reference for results raises questions. As far as I know, results can vary significantly depending on the target domain chosen.
5. Given that it is already the end of 2024, the paper lacks references to more recent works, including many methods from the DomainBed library, which are not mentioned.
6. It would be interesting to investigate how downstream fine-tuning of pre-trained models performs when combined with some out-of-distribution (OOD) algorithms, such as CLIPOOD or ADRMX.
7. The article's presentation lacks clarity, with no unified conclusions. Additionally, some labels in the figures, such as Figure 1, are not explained.

---

> ### Author Response · Authors · 2025-01-31
> **Response to reviewer AqLf**
>
> We sincerely appreciate your time and effort in reviewing our manuscript and providing insightful feedback. Your constructive comments have been invaluable, and we have carefully addressed each of them in our revised version. Below, we provide our detailed responses to your comments.
>
> __(W1) Limited contribution to academia:__
>
> We respectfully disagree with the view that our contribution to academia is relatively limited.
>
> Previous studies [1] have established scaling laws for in-distribution accuracy, contributing empirically validated generalizable insights to both the academic and industrial communities.
> Similarly, our findings demonstrate scaling laws in the context of OOD accuracy and calibration, providing guidance for the development of robust and reliable models, moreover, norm of fair comparison baseline for machine learning research community.
>
>
> __(W2) Newer architectures:__
>
> In this study, we have already conducted experiments using CLIP as a representative example of a Vision-Language Model, with the results presented in Appendix D.7.
>
> Given the constraints on computational resources, we prioritized experiments using widely adopted models, namely convolutional neural network, vision-transformer architecture etc.
>
> We leveraged the `timm` library to perform a comprehensive analysis using a diverse set of pre-trained models. However, publicly available pre-trained models for Vision-Language Models (VLMs) remain limited, unlike the extensive collection available in timm. This lack of available models makes it difficult to conduct a precise and thorough analysis.
> Therefore, within the constraints of our available resources, we have designed our experiments to ensure the most reliable and rigorous analysis possible.
>
> __(W3) The selection of datasets lacks diversity:__
>
> In this study, we have ensured a certain level of dataset diversity by incorporating datasets such as WILDS Camelyon and DomainBed, which cover a wide range of domains.
>
> To further supplement our validation on diverse datasets, we have added evaluation results on TerraIncognita in the Appendix D.2.
>
> Regarding your suggestion to include WOODS, we note that it primarily consists of time-series datasets, which fall outside the scope of this study (limited to image data). Additionally, significant differences in model architectures and evaluation methods make direct comparison within our framework challenging. Therefore, instead of incorporating WOODS, we have explicitly stated in the Abstract, Introduction, and Conclusion that this study is limited to image data.
>
>
> __(W4) Choice of target domain:__
>
> To efficiently conduct large-scale experiments, we adopted a design in which the first domain of each dataset is used as a reference, following the approach of [2].
> To eliminate potential biases from specific target domain selections, we conducted a consistent evaluation across five (six after rebuttal) different datasets.
>
> Additionally, we performed an extra validation on OfficeHome and PACS, analyzing the impact of choosing different target domains. The results of this analysis are presented in Appendix D.3.
>
> Ultimately, we confirmed that scaling trends remain consistent within the same architecture, reinforcing our original claims.
> As a result, while no clear trend was observed in cases where the environment was relatively easy to fit and model performance had saturated (e.g., “product” in OfficeHome and “photo” in PACS), in most cases, scaling trends remained consistent within the same model architecture, reinforcing our original claims.
>
> __(W5) Lack references to more recent works:__
>
> We have acknowledged relevant studies and added several additional references in our rebuttal. If there are any specific papers that are closely related to our work and should be cited, we would appreciate your suggestions.
>
> __(W6) Other OOD algorithms (e.g. CLIPOOD or ADRMX):__
>
> Investigating the combination of OOD algorithms such as CLIPOOD and ADRMX with fine-tuning of pre-trained models is indeed a fascinating research direction. However, as it extends beyond the scope of this study, we will consider it as a future research challenge.
>
> In particular, integrating new algorithms into our existing evaluation protocol while performing thorough hyperparameter tuning would require approximately 120,000 GPU hours, which translates to $367,200 on AWS V100 (P3.8xlarge). Given the substantial computational demands and the need for a well-structured research plan, we have opted not to pursue this investigation within the current study.
>
> [1] Kaplan, Jared, et al. "Scaling laws for neural language models." arXiv preprint arXiv:2001.08361 (2020).
>
> [2] Hiroki Naganuma, Kartik Ahuja, Shiro Takagi, Tetsuya Motokawa, Rio Yokota, Kohta Ishikawa, Ikuro Sato, Ioannis Mitliagkas, "Empirical Study on Optimizer Selection for Out-of-Distribution Generalizations", Transactions on Machine Learning Research (TMLR), 2023.

---

> ### Author Response · Authors · 2025-01-31
> **Response to reviewer AqLf**
>
> __(W7) No unified conclusions and clarity of Figure 1's label:__
>
> Thank you for your feedback. To clarify the conclusion, we have revised the paper (see Discussion and Conclusion).
>
> Regarding the legend labels in the figures (including Figure 1), we intentionally omitted certain labels to conserve space in the manuscript. However, the omitted labels are referenced in the captions, where we have highlighted the corresponding labels in bold. To further prevent confusion, we have now updated the manuscript to ensure these references are consistently formatted in bold.

---

### Review · Reviewer_AK7q · 2025-01-22

**Summary Of Contributions:**

*Idea*: The paper presents a systematic study of pre-trained model selection for fine-tuning OOD data. The scope is to gain insights on which pre-trained models to choose (especially which size), which pre-training data are the best, and which training strategy to use. In addition, the authors report both the accuracy and the ECE to gain better insights both on the performance and the calibration. The study is focused on computer vision models and data.

*Method*: The authors conduct experiments with 100 models, 5 pre-training datasets (variations of ImageNet), 5 data augmentation techniques, and report performance on 5 OOD datasets from DomainBed and WILD benchmarks. The code is implemented following the official implementations of the different algorithms and mostly follows the DomainBed framework.

*Experiments*: The authors perform large-scale experiments and provide guidelines for practitioners. In summary, they report that larger pre-training models and larger pre-training datasets improve both the performance and the calibration in OOD generalization.

**Audience:**

Yes

**Claims And Evidence:**

Yes

**Requested Changes:**

The following reference is present in double in the bibliography:
- Martin Arjovsky, Léon Bottou, Ishaan Gulrajani, and David Lopez-Paz. Invariant risk minimization.
arXiv preprint arXiv:1907.02893, 2019b.

Could the authors update that?

**Strengths And Weaknesses:**

**Strengths**
- Extensive and systematic experiments with clear practical takeaways
- Bridge a gap in the literature by jointly studying the models, the pre-training datasets, and training strategies
- Consider both performance and calibration for a fine-grained analysis of the results
- Provide scaling laws and phenomena common to many architectures
- Variety in the models and data to better cover the possible choice of practitioners
- Very well written with many experiments and ablation studies to ensure impactful insights

**Weaknesses**
Overall the paper is very well written and I believe it provides a very strong contribution to model selection for OOD generalization in computer vision. I list below some questions that are not weaknesses per se but could strengthen how future work relies on the current submission.
-I think that only conducting experiments on 5 OOD datasets is too small. Could the authors include more?
- This study is focused on only one modality (although it is a very broad-scope work). Do the authors have any idea of how their practical insights could translate to other modalities?
- The ECE metric has some flaws and other calibration metrics have been proposed in the literature [1], e.g., e Adaptive Calibration Error (ACE). Do the authors have any idea of how the results would translate using the ACE (or others) instead of the ECE?

**Questions**
- Do the authors have any insights from their work on how to design better (performance-wise) pre-trained models/training strategies or pre-training corpus of data?
- Same question to improve calibration?
- I think the term "uncertainty calibration" is unusual. Do the authors mean "uncertainty quantification" and 'calibration'?

[1] Measuring Calibration in Deep Learning, Nixon et al., 2020

---

> ### Author Response · Authors · 2025-01-31
> **Response to reviewer AK7q**
>
> Thank you for taking the time to review our manuscript and providing valuable feedback. We appreciate your constructive comments and have addressed each of them in our revised manuscript. Please find below our point-by-point response to your comments.
>
> __(W1) conducting experiments on 5 OOD datasets is too small:__
>
> We understand the importance of conducting experiments with additional OOD datasets. However, the objective of our study is not merely to evaluate performance across a large number of datasets but rather to comprehensively assess 100 pre-trained models—each with different training strategies, architectures, parameter sizes, and pre-training data—within the constraints of limited computational resources, while ensuring the reliability of our findings.
>
> In practice, adding even a single new dataset would incur approximately 24,000 GPU hours in additional costs under our existing evaluation protocol (equivalent to $73,440 on AWS V100 (P3.8xlarge)), which exceeds our feasible resource limitations.
> Moreover, models like ViT can exhibit lower performance than AlexNet if not properly tuned. However, interpreting such results at face value would be inappropriate. Instead of increasing the number of datasets, we prioritized evaluating a diverse set of well-tuned models on existing five different datasets to identify consistent trends.
>
> Our experimental results demonstrate that our claims (see Contributions) are consistently supported across five different OOD datasets. Therefore, our conclusions are not specific to a single dataset but rather hold more generally.
>
> In summary, our focus was not on evaluating as many datasets as possible but on ensuring a sufficiently representative and minimal set of datasets for comprehensive model evaluation while maintaining the reliability of our conclusions. In this regard, we believe our approach is well-justified.
>
> __(W2) limited modality:__
>
> While our study is specifically focused on the vision modality, and evaluations involving other modalities were outside our scope as we mentioned in (W1 - the priority of our research),  we conducted an evaluation using CLIP, a vision-language model, and the results are presented in Appendix D.7.
>
> Regarding the scope of our research, we acknowledge that the original manuscript did not explicitly mention this, which may have caused some confusion. To clarify, we have revised the abstract, introduction, and conclusion accordingly.
>
> __(W3) ACE vs ECE:__
>
> As you pointed out, our evaluation of ECE used equal-width binning, which we agree [1] claimed some limitations in certain cases.
> To address this, we comprehensively compared both metrics of ECE and ACE on our experimental protocol (see figure D.1 and D.2).
> Our results showed no significant difference in trends between ECE and ACE, leading to the same conclusions. This further reinforces the observations we initially reported.
>
> At the time of our paper submission, the optimal hyperparameters had already been identified by extensive hyperparameter search, allowing us to somewhat reduce the computational cost of recalculating ACE. However, this process still required significant resources—approximately 4,000 GPU hours in total, given 100 models × 5 datasets × approx 8 hours. With our 4-GPU cluster, this computation took approximately one and a half months.
>
> [1] Jeremy Nixon, Michael W Dusenberry, Linchuan Zhang, Ghassen Jerfel, and Dustin Tran. Measuring
> calibration in deep learning. In CVPR workshops, volume 2, 2019.

---

> > ### Comment · Reviewer_AK7q · 2025-02-03
> > **Thanks for the clarifications**
> >
> > I thank the authors for their clarifications.
> >
> > Regarding the number of OOD datasets, I better understand the authors' motivations (study of pre-training **extensively** rather than many test datasets) and understand the computational limitations. The author's answer is convincing and IMHO, could even appear in a discussion/limitation section (e.g., "future work could focus on extensive OOD evaluation but this is beyond the scope of the current work")
> >
> > I appreciate the authors' rephrasing regarding the modality. I did not point that as a limitation but was genuinely interested in the authors' takes regarding the generalization of their results to other types of data.
> >
> > I thank the author for the additional comparison with ACE which is satisfying as well as the correction of the unclear terms regarding calibration.
> >
> > Overall the paper is well written and I believe it provides a very strong contribution to model selection for OOD generalization in computer vision. I appreciate the authors' efforts to clarify the weaknesses/questions raised in my review.
> >
> > I maintain my acceptance proposal of this work.

---

> ### Author Response · Authors · 2025-01-31
> **Response to reviewer AK7q**
>
> __(Q1 and Q2) insights from our work on how to design better:__
>
> Many publicly available models are trained on ImageNet. However, our evaluation suggests that excessive fitting to ImageNet, may negatively impact OOD performance (see Fig. 4, 5). Therefore, pretraining strategies that avoid overfitting to a specific dataset may be preferable.
>
> Regarding calibration, we have observed scaling laws for the first time. These findings suggest that pre-training with a larger number of parameters and a large number of data is important for improving calibration.
>
> __(Q3) term of uncertainty calibration__:
>
> As you correctly pointed out, the term uncertainty calibration may cause confusion with uncertainty quantification and is therefore inappropriate. Originally, we used uncertainty calibration to refer to the proper adjustment of a model’s confidence. To clarify this, we have replaced uncertainty calibration with confidence calibration throughout the text.

---

### Decision · Action_Editor_Zyw4 · 2025-04-27

**Recommendation:** Accept as is

**Comment:**

This submission performs a relatively extensive empirical study of how to select the best pre-trained model for downstream transfer, enabling OOD generalization (in terms of accuracy) and calibration (in terms of ECE). The study is relevant and will be intriguing some different thoughts in the foundation model community. In particular, the joint consideration of generalization and calibration is new. Considering these aspects, the paper is recommended for (weak) acceptance.

**Audience:**

Yes, the community within TMLR focusing on OOD generalization and transfer from pre-trained models will be interested in this submission.

**Claims And Evidence:**

Most of the claims made in the submission are supported by empirical evaluation.